# A systematic review and meta-analysis of thigmotactic behaviour in the open field test in rodent models associated with persistent pain

**Xue Ying Zhang**[1], **Marta Diaz-delCastillo**[2], **Lingsi Kong**[3], **Natasha Daniels**[4], **William MacIntosh-Smith**[5], **Aya Abdallah**[5], **Dominik Domanski**[5], **Denis Sofrenovic**[6], **Tsz Pui (Skel) Yeung**[6], **Diego Valiente**[7], **Jan Vollert**[1], **Emily Sena**[8], **Andrew S. Rice**[1], **Nadia Soliman**[1] *

1 Pain Research, Department of Surgery and Cancer, Imperial College London, United Kingdom, 2 Department of Forensic Medicine, University of Aarhus, Aarhus, Denmark, 3 Department of Women and Children's Health, Faculty of Life Science and Medicine, King's College London, London, United Kingdom, 4 Bart's Health NHS Trust Whipps Cross Hospital, London, United Kingdom, 5 School of Medicine, Medical Science and Nutrition, University of Aberdeen, Aberdeen, United Kingdom, 6 Faculty of Medicine, Imperial College London, London, United Kingdom, 7 Institute of Psychiatry, Psychology and Neuroscience, King's College London, London, United Kingdom, 8 Centre for Clinical Brain Sciences, University of Edinburgh, Edinburgh, United Kingdom

* n.soliman16@imperial.ac.uk

**Data Availability Statement:** The data underlying the results presented in the study are available from OSF at https://osf.io/rmt97/.

## Abstract

Thigmotaxis is an innate predator avoidance behaviour of rodents. To gain insight into how injury and disease models, and analgesic drug treatments affect thigmotaxis, we performed a systematic review and meta-analysis of studies that assessed thigmotaxis in the open field test. Systematic searches were conducted of 3 databases in October 2020, March and August 2022. Study design characteristics and experimental data were extracted and analysed using a random-effects meta-analysis. We also assessed the correlation between thigmotaxis and stimulus-evoked limb withdrawal. This review included the meta-analysis of 165 studies We report thigmotaxis was increased in injury and disease models associated with persistent pain and this increase was attenuated by analgesic drug treatments in both rat and mouse experiments. Its usefulness, however, may be limited in certain injury and disease models because our analysis suggested that thigmotaxis may be associated with the locomotor function. We also conducted subgroup analyses and meta-regression, but our findings on sources of heterogeneity are inconclusive because analyses were limited by insufficient available data. It was difficult to assess internal validity because reporting of methodological quality measures was poor, therefore, the studies have an unclear risk of bias. The correlation between time in the centre (type of a thigmotactic metric) and types of stimulus-evoked limb withdrawal was inconsistent. Therefore, stimulus-evoked and ethologically relevant behavioural paradigms should be viewed as two separate entities as they are conceptually and methodologically different from each other.

**Funding:** XYZ was in receipt of funding from the Medical Research Council (grant number MR/N014103/1) and NS funded by the Jennie Gwynn Legacy Fund. The funders had no role in study design, data collection and analysis, decision to publish, or preparation of the manuscript. The remaining author received no specific funding for this work.

**Competing interests:** The authors have declared that no competing interests exist.

## 1. Introduction

Chronic pain is a leading cause of the global disease burden [1]. However, current analgesic drugs lack efficacy and have unwanted side effects, leaving chronic pain patients with inadequate pain relief. Thus, there is a need for analgesics with better efficacy and safety profiles [1]. Several promising novel drugs with encouraging preclinical results have failed to exhibit clinical efficacy [2–6]. This translational failure has led researchers to question the validity of current animal pain associated models, in particular how well they mimic the disease and whether the outcome measures used are measuring the intended clinical construct.

Surrogate outcome measures are used to assess pain-like behaviours in animal models. Several large systematic reviews and reviews of historical data have demonstrated that stimulus-evoked limb withdrawal is the most frequently used type of behavioural outcome measure in preclinical pain research However, stimulus-evoked paradigms have limitations that can undermine their validity. First, they have limited clinical relevance [4]. They are involuntary responses triggered by spinal reflexes so they cannot address spontaneous pain and clinical phenotypes relating to sensory loss of function, and for example, patients with neuropathic pain experience both phenotypes [7–9]. Second, information regarding the affective and physical dimensions of pain cannot be addressed, and these factors are usually emphasised in clinical settings [7]. Moreover, rodents can also learn to associate premature withdrawal with less stimulations and human interaction [10]. Lastly, they cannot distinguish analgesic effects from sedation, which can further compromise their predictive validity. Therefore, animal pain research needs to identify and validate alternative outcome measures that can address these limitations.

Various voluntary and operant behaviours have been reported to be affected by experimental nociception [11–13]. These complex ethologically relevant behaviours are increasingly being exploited to assess the impact of persistent and spontaneous pain on an animal's physical function and general wellbeing [14–18]. It should be noted that these ethologically relevant behaviours are not pain specific, so it is crucial to contextualise them to pain research. Pain contextualisation can be achieved by showing that changes in these behaviours are caused by injury and disease models associated with pain and that changes can be reversed by administering known analgesics.

Thigmotaxis is an innate predator avoidance rodent behaviour that is usually displayed when they are under stress [19], and it can be objectively measured in an open field test (OFT). The behaviour is characterised by the preference of a rodent to seek shelter instead of exposing itself to the aversive open area i.e., an stressed animal is more likely to stay in proximity of the walls and avoid the relatively exposed centre of the open area and is less likely to explore the open area/novel environment than a less stressed animal [19]. This behavioural paradigm is commonly used for animal research of psychiatric disorders, particularly anxiety, but its use in the pain field is relatively novel. It has been postulated that the behaviour could give insight into the affective, cognitive, and sensory dysfunction associated with pain. It could also reflect the clinical observations of exacerbated avoiding behaviours and anxiodepressive disorders that are associated with patients with chronic pain [20–22]. Existing studies have shown increased thigmotaxis in several rodent experimental models associated with chronic pain [23–26]. Furthermore, thigmotaxis has shown sensitivity towards clinically approved analgesics where the behaviour was reduced in rodents with experimental chronic pain by gabapentin and morphine [23, 27, 28]. Moreover, the motor function of an animal can also be assessed during the OFT, therefore thigmotaxis is a measurable construct that can potentially address aspects of the affective and physical dimensions of pain.

However, like other ethologically relevant behaviours, thigmotaxis can be perturbed by various factors. For example, thigmotaxis can be enhanced by increasing light intensity in the open area [29]. The reliability of thigmotaxis as a pain-related outcome measure across different disease models is also unclear. To improve our understanding, we conducted a systematic review accompanied with a meta-analysis to assess the strengths and limitations of using thigmotaxis during the OFT in animal pain research. We have appraised the current experimental designs including the OF apparatus, type of thigmotactic outcome measures and experimental conditions, and have tried to identify factors that can influence the final thigmotactic outcome. In addition, we have assessed the correlation between the widely used stimulus-evoked limb withdrawal and thigmotactic outcomes to aid our assessment of the translational value of these preclinical models.

### 1.1. Aims and objectives

This systematic review aimed to 1) assess whether thigmotaxis can be affected by injury and disease models associated with persistent pain and analgesic drug treatments in rodents; 2) explore study design characteristics and assess their influence on thigmotactic outcomes; 3) perform a risk of bias assessment to evaluate and assess their impact on thigmotactic outcomes; 4) identify the presence of publication bias and determine its direction and magnitude; 5) assess the correlation of thigmotaxis and total distance travelled and stimulus-evoked limb withdrawal in the same cohorts of animals.

## 2. Results

### 2.1. Study selection

A total of 1819 publications were retrieved from three systematic searches; of which, 705 were included after title and abstract screening. Full-text screening identified 181 studies that met the inclusion criteria (Fig 1). A total of 16 studies were not included for meta-analysis as they reported no data for the meta-analysis despite contacting the authors for the missing data. Thus, data of 165 studies were included in the meta-analysis [30].

### 2.2. Study characteristics

The 181 studies included a total of 5998 rodents (3943 in animal modelling and 2055 in drug experiments). Thigmotaxis was reported in 66 different rodent models associated with persistent pain. These models are listed according to the classification (28 model classes) in Table 1; nerve injury induced neuropathy (40%, k = 147) and inflammation (13%, k = 49) were the most frequently reported. Eighty drugs are classified by their mechanism of action (53 drug classes) using the IUPHAR/BPS guide to Pharmacology (https://www.guidetopharmacology.org/) as listed in Table 2; gabapentinoids (10%, k = 15) were the most frequently investigated. Rats were used in 52% (k = 193) of the experiments and 48% (k = 178) used mice. Furthermore, 81% (k = 303) used male animals, 16% (k = 59) used female animals, 2% (k = 7) used mixed sexes, and 1% (k = 2) did not report the sex of the animals used. Information regarding acclimatisation, animal husbandry and experimental conditions is summarised in S1 File. The reporting of humidity, noise and vibration level in both housing and testing room were low. For acclimatisation and animal husbandry, number of light/dark hours was the most frequently reported (85% studies), with a range between 12/12 hr to 14/10 hr and a median of 12/12hr. For experimental conditions, shape of the open arena was the most frequently reported (96% studies), and most studies reported using a rectangular or square shaped open arena.

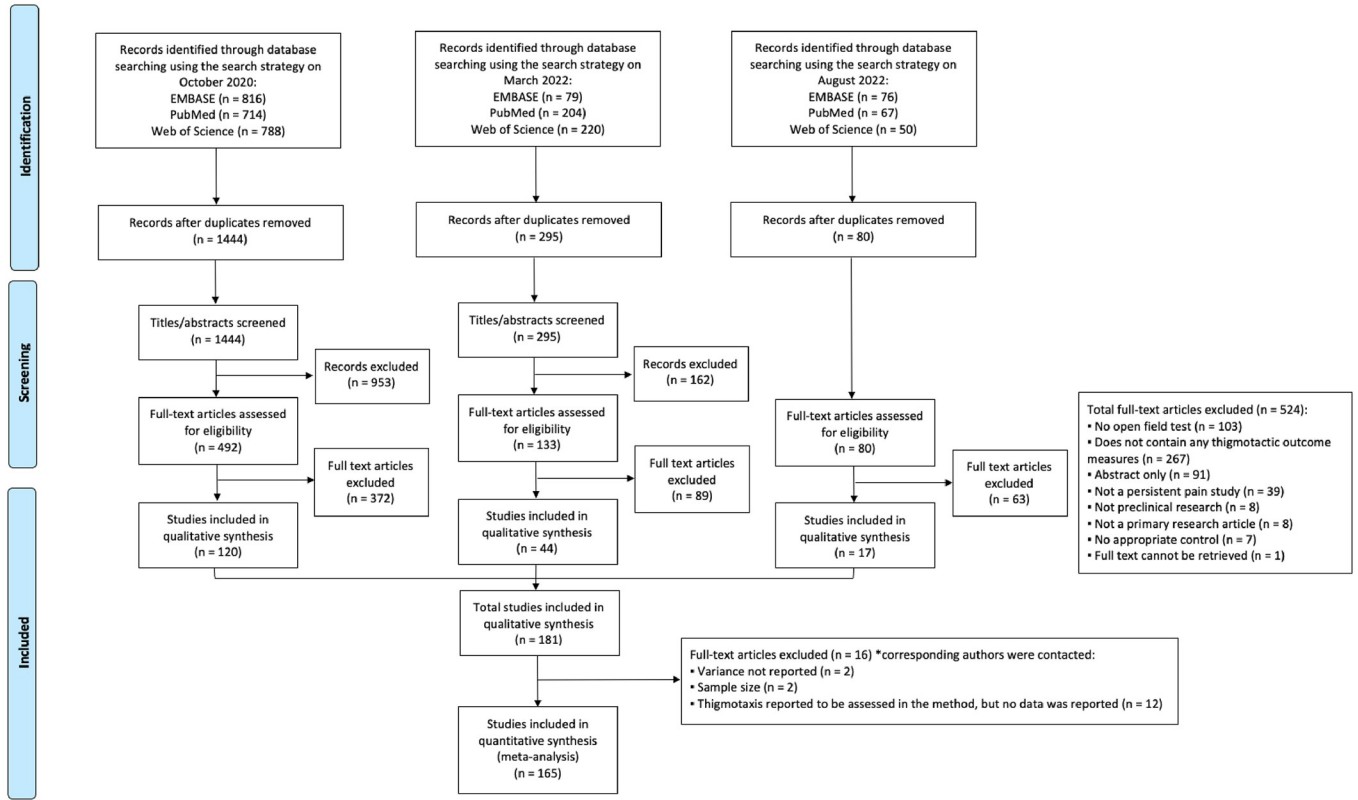

**Fig 1. PRISMA flow diagram.** A flow diagram of publications identified through three separate systematic searches of three electronic databases (EMBASE, PubMed, and Web of Science). The diagram illustrates the number of records (n) at deduplication, screening, and study eligibility for both qualitative and quantitative analyses. Reported in accordance with the PRISMA 2020 guideline [27].

## 2.3. Meta-analysis of thigmotactic outcomes

Species accounted for a significant proportion of heterogeneity (animal modelling: Q = 23.27, df = 1, $p < 0.0001$). Thus, animal modelling and drug treatment data of rats and mice were analysed separately. The type of control used in animal modelling experiments (i.e. sham vs naïve) did not account for a significant proportion of heterogeneity (rats: Q = 0.71, df = 1, $p = 0.4$; mice: Q = 3.13, df = 1, $p = 0.08$). There are fewer comparisons using naïve controls (rats: sham = 91 vs naïve = 12; mice: sham = 116 vs naïve = 5), and due to the known differences between the two control types, they have been analysed separately. Thigmotaxis was increased by injury and disease models associated with persistent pain compared to naïve controls in rats, the full analysis is presented in S2 File. There were too few mouse experiments using naïve controls to analyse how thigmotaxis can be affected by injury and disease models associated with persistent pain (S2 File). We have divided the reporting of results by type of study (i.e., modelling experiments or drug intervention experiments) and by species (i.e., rats and mice) therefore we have four datasets: (i) dataset 1 –effects of modelling persistent pain on thigmotaxis outcomes in rats, (ii) dataset 2—effects of analgesic drug interventions on thigmotaxis outcomes in rat injury and disease models associated with persistent pain, (iii) dataset 3 – effects of modelling persistent pain on thigmotaxis outcomes in mice, (iv) dataset 4 –effects of analgesic drug interventions on thigmotaxis in mouse injury and disease models associated with persistent pain.

**Table 1. Summary of the model types used in animal modelling and drug treatment experiments during the OFT for assessing thigmotaxis.**

| Model Type (No. of total studies) | Model Name | Thigmotaxis | | | |
|---|---|---|---|---|---|
| | | No. of studies | No. of cohort-level comparisons | No. of rats | No. of mice |
| Neuropathy–nerve injury (83) | Sciatic nerve ligation | 31 | 53 | 395 | 363 |
| | Spared nerve injury | 23 | 47 | 365 | 423 |
| | Spinal nerve ligation | 10 | 22 | 349 | 79 |
| | Partial sciatic nerve ligation | 6 | 6 | 86 | 56 |
| | Infraorbital nerve constriction | 5 | 8 | 68 | 93 |
| | Spinal nerve transection | 4 | 6 | 65 | 36 |
| | Chronic compression of multiple dorsal root ganglia | 1 | 2 | 38 | - |
| | Partial transection of the infraorbital nerve | 1 | 1 | - | 42 |
| | Sciatic nerve resection | 1 | 1 | 18 | - |
| | Ligation of the common peroneal nerve | 1 | 1 | - | 14 |
| Inflammation (19) | Complete Freund's Adjuvant (non-articular) | 17 | 42 | 146 | 447 |
| | Lipopolysaccharide (LPS) | 1 | 6 | 59 | - |
| | Bee venom | 1 | 1 | 20 | - |
| Migraine (12) | Nitroglycerine | 5 | 10 | 114 | - |
| | Stress induced migraine | 1 | 4 | - | 64 |
| | Optogenetic spreading depression | 1 | 2 | - | 123 |
| | AAV2-EF1a-DIO-ChR2(E123A)-mCherry (activate CGRP neurons in the medial nucleus) | 1 | 2 | - | 39 |
| | Calcitonin gene-related peptide | 1 | 2 | - | 22 |
| | Prostaglandin E2, histamine, serotonin and bradykinin (intracranial) | 1 | 1 | 19 | - |
| | LEI106 | 1 | 1 | 16 | - |
| | Acrolein | 1 | 1 | 15 | - |
| Visceral inflammation (11) | 2,4,6-trinitrobenzene sulfonic acid induced colitis | 5 | 8 | 157 | - |
| | Dextran sulfate sodium induced colitis | 2 | 3 | - | 74 |
| | Zymosan induced colitis | 1 | 8 | - | 72 |
| | Turpentine induced bladder inflammation | 1 | 2 | 45 | - |
| | Cerulein induced pancreatitis | 1 | 1 | - | 20 |
| | Deoxycholic acid induced colonic inflammation | 1 | 1 | 16 | - |
| Spinal cord injury (9) | Spinal cord injury | 9 | 26 | 81 | 323 |
| Arthropathy (5) | Monosodium iodoacetate (intra-articular) | 2 | 3 | 20 | 30 |
| | Complete Freund's Adjuvant (intra-articular) | 1 | 5 | 49 | - |
| | Carrageenan (knee joint cavity) | 1 | 2 | 30 | - |
| | Kaolin and carrageenan induced monoarthritis (synovial cavity) | 1 | 1 | 12 | - |
| Neuropathy–antiretroviral therapy (5) | Dideoxycytidine (ddC) | 3 | 8 | 135 | - |
| | Stavudine (D4T) | 1 | 1 | 24 | - |
| | Indinavir | 1 | 1 | 16 | - |
| Neuropathy–chemotherapy (5) | Paclitaxel | 3 | 6 | 19 | 72 |
| | Cyclophosphamide | 1 | 2 | 18 | - |
| | A mixture of FOLFOX components (oxaliplatin, 5-FU, LV calcium salt) | 1 | 1 | 20 | - |
| Bone fracture (4) | Tibial fracture | 4 | 5 | - | 106 |
| Diabetic neuropathy (4) | Streptozotocin | 4 | 9 | 81 | 20 |

(*Continued*)

**Table 1.** (Continued)

| Model Type (No. of total studies) | Model Name | Thigmotaxis | | | |
|---|---|---|---|---|---|
| | | No. of studies | No. of cohort-level comparisons | No. of rats | No. of mice |
| Post-surgical pain (4) | Ovariectomy | 2 | 2 | 9 | 11 |
| | Paw surgical incision | 1 | 4 | 44 | - |
| | Rib retraction via thoracotomy | 1 | 2 | 32 | - |
| Cancer (3) | B16-F10 murine melanoma cell | 1 | 9 | - | 128 |
| | B6 Hi-Myc | 1 | 1 | - | 20 |
| | MRMT-1/Luc cells | 1 | 1 | 21 | - |
| Fibromyalgia (3) | Reserpine induced fibromyalgia | 2 | 2 | - | 22 |
| | Acid injection | 1 | 3 | 60 | - |
| Parkinson's Disease (3) | MPTP induced | 1 | 2 | - | 41 |
| | A53T transgenic mice | 1 | 1 | - | 24 |
| | 6-OHDA | 1 | 1 | - | 24 |
| Traumatic brain injury (2) | Closed head traumatic injury | 2 | 4 | 50 | 11 |
| Combined models (2) | Tibial fracture and closed-head traumatic brain injury | 1 | 2 | - | 22 |
| | Sciatic nerve ligation and ovariectomy | 1 | 1 | 9 | - |
| Acute herpes Zoster (1) | Varicella zoster virus inoculation | 1 | 2 | 34 | - |
| Autoimmune encephalomyelitis (1) | MOG35-55; saponin Quil A; PTX | 1 | 2 | - | 46 |
| Hyperhomocysteinemia (1) | Methionine | 1 | 1 | 30 | - |
| Formalin (1) | Formalin | 1 | 2 | - | 32 |
| mGlu5R overexpression in the caudal part prelimbic area (1) | mGlu5G-expressed lentivirus | 1 | 1 | 20 | - |
| Myocardial infarction (1) | Myocardial infarction | 1 | 1 | 15 | - |
| Neuropathy–resiniferatoxin (1) | Resiniferatoxin | 1 | 2 | 48 | - |
| Neuropathy–HIV and antiretroviral therapy (1) | HIV gp120 + dideoxycytidine (ddC) | 1 | 2 | 32 | - |
| Neuropathy–HIV (1) | HIV gp120 | 1 | 1 | 16 | - |
| Neurotoxicity (1) | Diiodohydroxyquinoline | 1 | 4 | 80 | - |
| P2Y12 deficiency (1) | P2Y12 homozygous KO | 1 | 1 | - | 24 |
| Stroke (1) | Central post stroke pain | 1 | 5 | 79 | - |
| **TOTAL** | | | **371** | **3075** | **2923** |

**2.3.1. Dataset 1: Effects of modelling persistent pain on thigmotaxis outcomes in rat experiments using sham controls.** A total of 68 studies, containing 91 cohort-level comparisons, 1678 rats, and a sample size range from 6 to 60 with a median of 16 animals per group, assessed the effects of 21 types of disease models associated with persistent pain on thigmotaxis. Eighty percent (k = 73) of experiments assessed thigmotaxis by measuring time spent in the centre, 10% (k = 9) measured distance travelled in the centre, 7% (k = 6) measured entries to the centre, 2% (k = 2) measured time spent in the periphery and 1% (k = 1) measured number of central crossings.

*Thigmotaxis was increased in injury and disease models associated with persistent pain compared to sham controls in rats.* The model significantly increased thigmotaxis in rats compared to sham controls (SMD = -3.35 [95%CI -4.01 to -2.69]). Heterogeneity was high (Q = 4651.06, df = 90, $p < 0.0001$, $I^2 = 98\%$) (Fig 2).

**Table 2. Classification of the drug treatments.**

| Drug Class | Drug Name | Thigmotaxis | | | |
|---|---|---|---|---|---|
| | | No. of studies | No. of cohort-level comparisons | No. of rats | No. of mice |
| **Gabapentinoid (10)** | Gabapentin | 6 | 10 | 76 | 36 |
| | Pregabalin | 3 | 3 | 31 | 14 |
| | Mirogabalin besylate | 1 | 2 | 36 | - |
| **Hydrogen sulphide donor (7)** | Allyl isothiocyanate (A-ITC) | 2 | 2 | - | 31 |
| | Phenyl isothiocyanate (P-ITC) | 2 | 2 | - | 31 |
| | GYY4137 | 2 | 2 | - | 28 |
| | DADS (Diallyl disulphide) | 1 | 1 | - | 12 |
| **Unknown mechanism of action (6)** | Paeonia lactiflora | 1 | 3 | 33 | - |
| | 8-O-Acetyl Shanzhiside Methylester | 1 | 3 | - | 24 |
| | Shorea roxburghii polyphenol extract | 1 | 2 | 18 | - |
| | α-lipoic acid (antioxidant) | 1 | 1 | - | 16 |
| | Albiflorin | 1 | 1 | 12 | - |
| | Betanin (red beetroot extract) | 1 | 1 | - | 12 |
| **Serotonin-norepinephrine reuptake inhibitor (5)** | Duloxetine | 2 | 2 | 31 | - |
| | Fluoxetine | 2 | 2 | 21 | - |
| | Citalopram | 1 | 1 | 17 | - |
| **GABA agonist (4)** | Muscimol | 2 | 2 | 30 | 15 |
| | Diazepam | 2 | 2 | 31 | - |
| **Tetracycline antibiotic (4)** | Minocycline | 4 | 5 | 80 | - |
| **Opioid (4)** | Morphine | 3 | 7 | 66 | 30 |
| | DALDA (mu opioid agonist) | 1 | 2 | - | 23 |
| **PPAR (peroxisome proliferator-activator-α) antagonist (4)** | GW6471 | 1 | 1 | 10 | - |
| | GSK0660 | 1 | 1 | 10 | - |
| | GW9662 | 1 | 1 | 10 | - |
| | Resveratrol | 1 | 1 | 14 | - |
| **Tricyclic antidepressant (TCA) (4)** | Amitriptyline | 3 | 3 | 32 | 14 |
| | Imipramine | 1 | 1 | 13 | - |
| **Antimalarial (3)** | Artemether (AR-TN) [artemisinin derivative] | 1 | 1 | - | 14 |
| | Artesunate (ART) [artemisinin derivative] | 1 | 1 | - | 14 |
| | Dihydroartemisinine (DHA) [artemisinin derivative] | 1 | 1 | - | 14 |
| **Anti-TNF (TNF blocker) (2)** | Etanercept | 1 | 1 | - | 21 |
| | Xpro1595 | 1 | 1 | - | 21 |
| **Cannabinoid (2)** | Cannabidiol | 2 | 5 | 50 | - |
| **CGRP receptor antagonist (2)** | CGRP 8–37 | 1 | 2 | 72 | - |
| | Olcegepant (BIBN4096BS) | 1 | 2 | - | 32 |
| **Dopamine receptor agonist (2)** | Quinpirole | 1 | 1 | - | 20 |
| | SKF38393 | 1 | 1 | - | 19 |
| **Palmitoylethanolamide (2)** | L-29 (Palmitoylallylamide)– palmitoylethanolamide analogue | 1 | 1 | 20 | - |
| | N-palmitoylethanolamide | 1 | 1 | 10 | - |
| **mTOR inhibitor (2)** | Rapamycin | 2 | 2 | 38 | - |
| **MEK1 and MEK2 inhibitor (2)** | U0126 | 2 | 3 | 29 | - |

(*Continued*)

**Table 2.** (*Continued*)

| Drug Class | Drug Name | Thigmotaxis | | | |
|---|---|---|---|---|---|
| | | No. of studies | No. of cohort-level comparisons | No. of rats | No. of mice |
| **Translocator protein agonist (2)** | ZBD-2 | 1 | 2 | - | 24 |
| | AC-5216 | 1 | 1 | 15 | - |
| **TRPA1 antagonist (2)** | HC-030031 | 1 | 1 | - | 16 |
| | TRPA1 oligonucleotides antisense | 1 | 1 | - | 16 |
| **Combined therapies (2)** | Dexmedetomidine + 2'3'-cGAMP | 1 | 1 | 9 | - |
| | Ketamine + 2'3'-cGAMP | 1 | 1 | 9 | - |
| **5-HT receptor agonist (1)** | Serotonin | 1 | 1 | - | 20 |
| **Acetylcholinesterase inhibitor (1)** | Scopoletin | 1 | 4 | - | 66 |
| **Angiotensin II type 2 receptor agonist (1)** | Novokinin | 1 | 1 | 20 | - |
| **Angiotensin II type 2 receptor antagonist (1)** | EMA300 | 1 | 1 | 20 | - |
| **Adenylyl cyclase 1 inhibitor (1)** | NB001 | 1 | 4 | - | 36 |
| **FFA1 and retinoid X receptor-α agonist (1)** | Docosahexaenoic acid | 1 | 3 | 62 | - |
| **β adrenoceptor antagonist (1)** | Propranolol | 1 | 2 | 32 | - |
| **Flavone (1)** | Luteolin | 1 | 1 | - | 18 |
| **Cysteine protease (1)** | Bromelain | 1 | 4 | 40 | |
| **GABA$_A$ receptor positive allosteric modulator (1)** | Etifoxine | 1 | 1 | - | 20 |
| **GRP30 agonist (1)** | G1 | 1 | 3 | - | 24 |
| **HCN (hyperpolarization-activated cyclic nucleotide-gated) channel blocker (1)** | ZD7288 | 1 | 1 | - | 14 |
| **Heme oxygenase 1 inducer (1)** | Protoporphyrin | 1 | 1 | - | 16 |
| **Botulinum toxin (1)** | Botulinum toxin type A | 1 | 1 | - | 16 |
| **NMDA receptor antagonist (1)** | Ketamine | 1 | 2 | 25 | - |
| **$N^G$, $N^G$-Dimethylarginine dimethylaminohydrolase (DDAH) 1 inhibitor (1)** | N5-(1-imino-3-butenyl)-L-ornithine (L-VNIO) | 1 | 2 | - | 56 |
| **NPS receptor agonist (1)** | Neuropeptide S | 1 | 2 | 24 | - |
| **NSAID (1)** | Ibuprofen | 1 | 1 | 8 | - |
| **Orexin receptor agonist (1)** | Orexin-A | 1 | 2 | 18 | - |
| **Fatty acid derived specialised pro-resolving mediator (SPM) (1)** | Resolvin D5 | 1 | 3 | 27 | - |
| **PAC1 receptor antagonist (1)** | PACAP(6–38) | 1 | 1 | 10 | - |
| **Paracetamol (1)** | Paracetamol | 1 | 1 | 12 | - |
| **Protein kinase Mzeta inhibitor (1)** | Zeta interacting protein (ZIP) | 1 | 1 | 12 | - |
| **Selective α2-adrenergic receptor agonist (1)** | Dexmedetomidine | 1 | 1 | 9 | - |
| **Selective BK$_{ca}$ channel agonist (1)** | NS1619 | 1 | 1 | 12 | - |
| **Selective MEK1 pathway inhibitor (1)** | PD98059 | 1 | 1 | 10 | - |
| **Selective NLRP3 inhibitor (1)** | MCC950 | 1 | 1 | 12 | - |
| **FZD$_8$ negative allosteric modulator (tricyclic anticonvulsant) (1)** | Carbamazepine | 1 | 1 | 20 | - |
| **Sodium channel blocker** | Lidocaine | 1 | 1 | 24 | - |
| **Free fatty acid (FFA1) receptor agonist (1)** | 2-OHOA (2-Hydroxy Oleic Acid) | 1 | 1 | 12 | - |
| **Vasopressin and oxytocin receptor agonist (1)** | Oxytocin | 1 | 1 | - | 14 |

(*Continued*)

**Table 2.** (Continued)

| Drug Class | Drug Name | Thigmotaxis | | | |
| --- | --- | --- | --- | --- | --- |
| | | No. of studies | No. of cohort-level comparisons | No. of rats | No. of mice |
| **CYP4F2 inhibitor (1)** | Sesamin | 1 | 1 | - | 10 |
| **COMT (Catechol-O-methyltransferase) and MAO (monoamine oxidase) inhibitor (1)** | Rosmarinic acid | 1 | 1 | - | 16 |
| **TOTAL** | | | 147 | 1232 | 823 |

Sensitivity analysis showed that the removal of the two comparisons from Ren et al., 2021 (b) and Wu et al., 2017 which reported very large effect sizes did not significantly change the summary effect size (SMD = -3.05 [95%CI -3.57 to -2.53]).

*A positive correlation between time spent in the centre and total distance travelled in rat modelling experiments.* A total of 60 cohort-level comparisons assessed both time in the centre and total distance travelled in rat modelling experiments. Overall, total distance travelled was significantly reduced by injuries and disease models associated with persistent pain (SMD = -2.04 [95%CI -2.86 to -1.22]) (Fig 3). Total distance travelled was reduced most in neurotoxicity models (SMD = -6.30, [95%CI -6.37 to -6.23], k = 4) and was reduced least in diabetic neuropathy models (SMD = 0.14, [95%CI -24.35 to 24.63], k = 2).

There is a moderate positive correlation (Coefficient = 0.46, df = 58, $p$ = 0.0002) (Fig 4) between time spent in the centre and total distance travelled. Where there were enough cohort-level comparisons, we also assessed the correlation between time in the centre and total distance travelled within the same model class. In rats modelled with nerve injury induced neuropathy (k = 24), time in the centre did not correlate with total distance travelled (Coefficient = 0.22, df = 22, $p$ = 0.31) (Fig 5).

*Effects of animal model and animal characteristics on the thigmotactic outcome in rat modelling experiments.* Visceral inflammation (k = 11) and nerve injury induced neuropathy (k = 39) were the only model classes with enough comparisons for stratified meta-analysis. The model class did not account for a significant proportion of heterogeneity (Q = 0.17, df = 1, $p$ = 0.68) (Fig 6A). Sensitivity analysis showed that the removal of the comparison from Ren et al., 2021 (b) significantly affected the summary effect size of visceral inflammation (SMD = -2.44 [95%CI -3.94 to -0.95]).

Male rats were used in 82% (k = 75) of experiments, female rats were used in 17% (k = 15), and 1% (k = 1) did not report the sex of the rats used. Sex accounted for a significant proportion of heterogeneity (Q = 15.59, df = 1, $p < 0.0001$) (Fig 6B). Thigmotaxis was more significantly increased in male rats.

Four rat strains were reported. Sprague-Dawley was the most reported strain (66%, k = 60). Wistar and Sprague-Dawley rats were the only two strains with enough comparisons for stratified meta-analysis. Strain did not account for a significant proportion of heterogeneity (Q = 0.05, df = 1, $p$ = 0.82) (Fig 6C).

Open field tests were conducted from immediately after model induction to 35 days after nerve injury (for which there were 41 comparisons that reported time since model induction). Time accounted for a significant proportion of heterogeneity (Q = 652.50; d.f. = 13; p <0.0001) (Fig 7A). However, this was not the case when time was analysed in weekly blocks (Q = 3.31; d.f. = 3; p = 0.35) (Fig 7B).

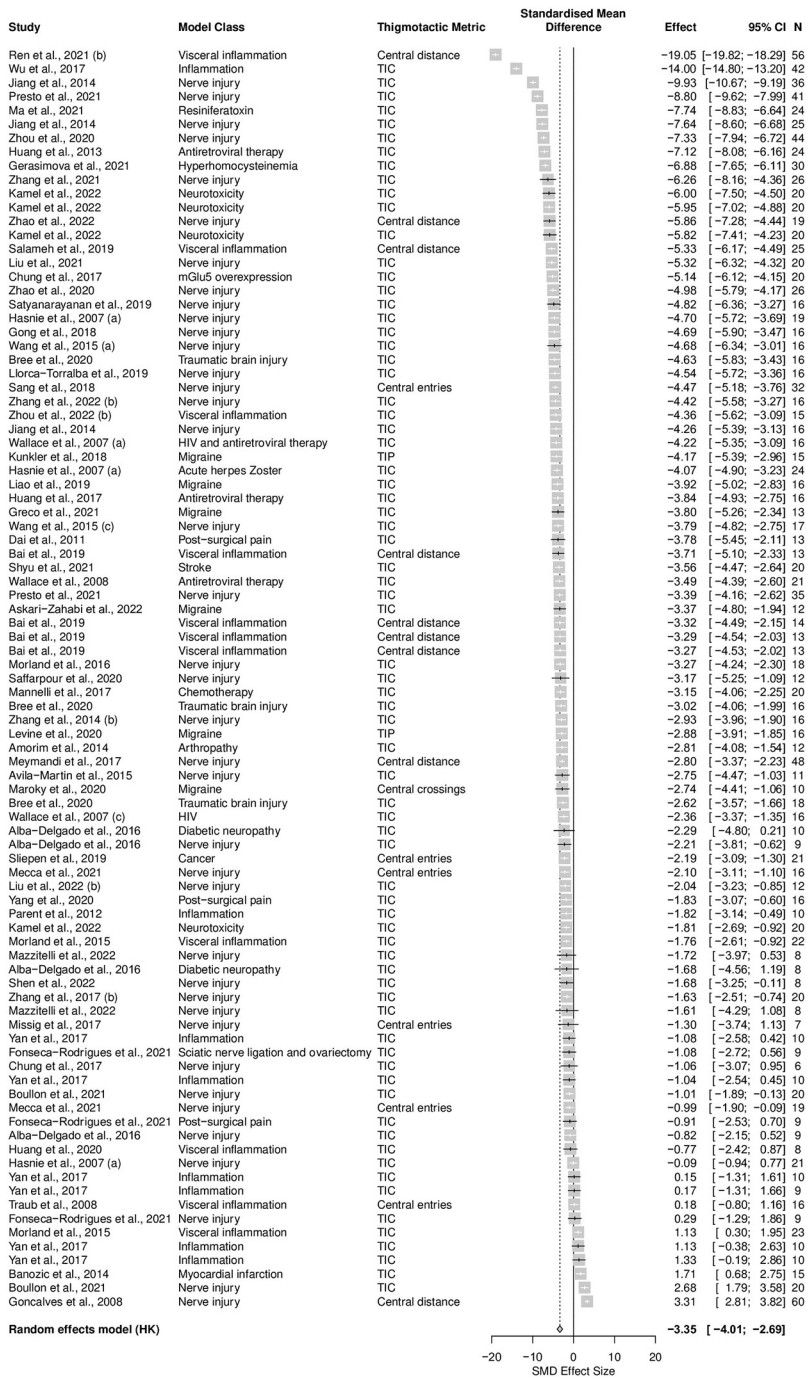

**Fig 2. Forest plot of effect of modelling in rats.** A summary forest plot of the 91 cohort-level comparisons which assessed the impact of modelling compared to sham control on rat thigmotaxis. For each comparison, an effect size was calculated using the Hedges' *g* SMD method. Effect sizes were pooled using the random effects model. The restricted maximum-likelihood method was used to estimate heterogeneity. The overall effect size is -3.35 [95% CI -4.01 to -2.69]; Q = 4651.06, $df$ = 90, $p < 0.0001$, $I^2$ = 98%. The size of the square represents the weight, which reflects the contribution of each comparison with the pooled effect estimate. TIC, time spent in the centre; TIP, time spent in the periphery. CI, confidence interval; N, number of animals.

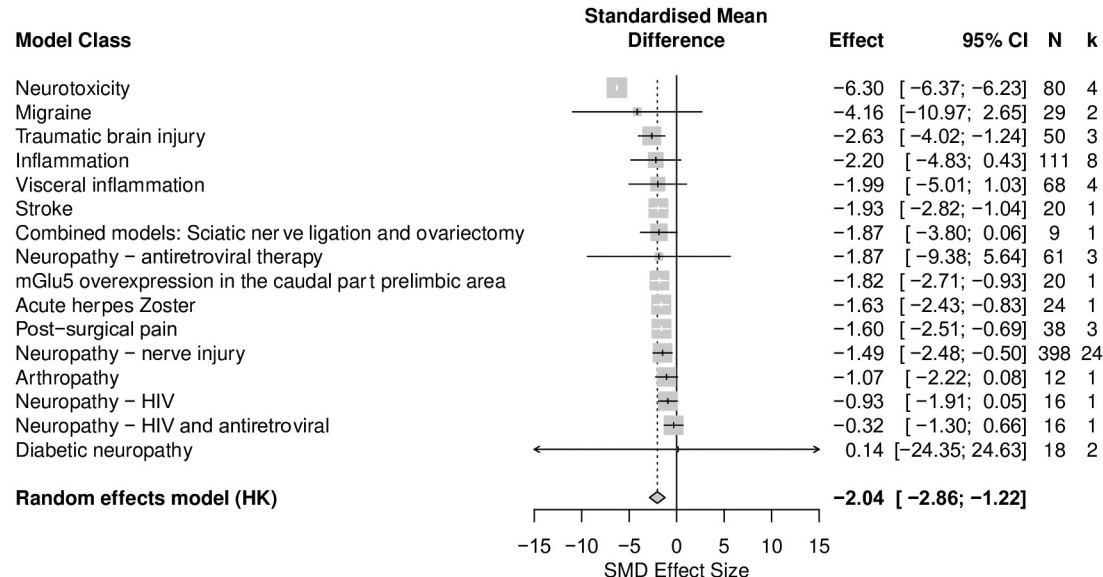

**Fig 3. Forest plot of impact of modelling on total distance travelled in rats.** Sixty cohort-level comparisons assessed the impact of modelling on total distance travelled in rats. The overall effect size is -2.04 [95% CI -2.86 to -1.22]. The size of the square represents the weight, which reflects the contribution of each comparison with the pooled effect estimate. CI, confidence interval; k, number of cohort-level comparisons; N, number of animals.

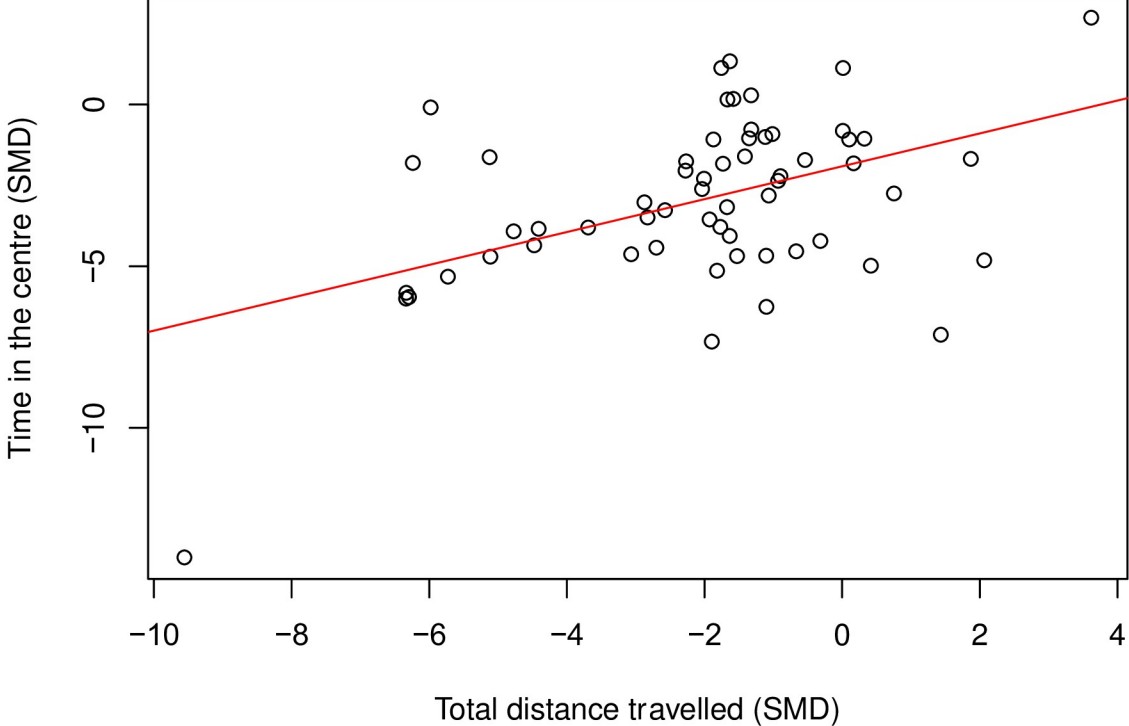

**Fig 4. A Pearson's correlation test between time in the centre and total distance travelled in rat modelling experiments.** There is a moderate positive correlation (Coefficient = 0.46, df = 58, p = 0.0002, k = 60). A line of best fit (in red) was drawn. SMD, standardised mean difference.

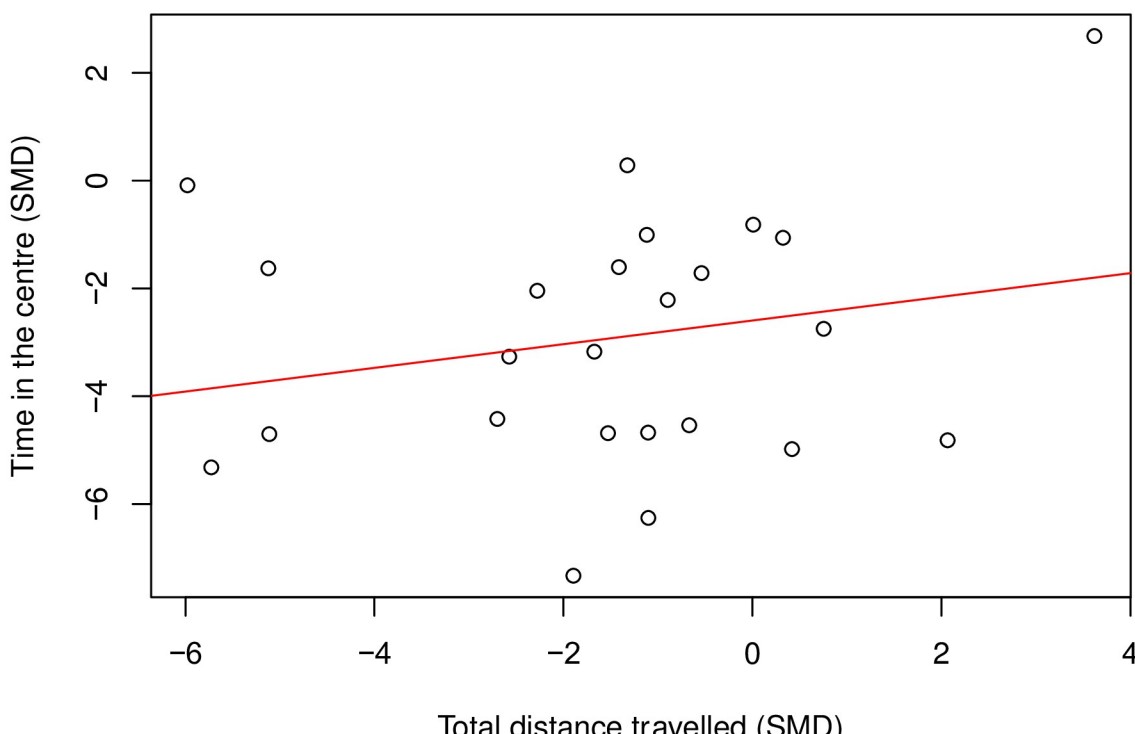

**Fig 5. A Pearson's correlation test between time in the centre and total distance travelled in rats modelled with nerve injury.**
There is no correlation (Coefficient = 0.22, df = 22, $p$ = 0.31, k = 24). A line of best fit (in red) was drawn. SMD, standardised mean difference.

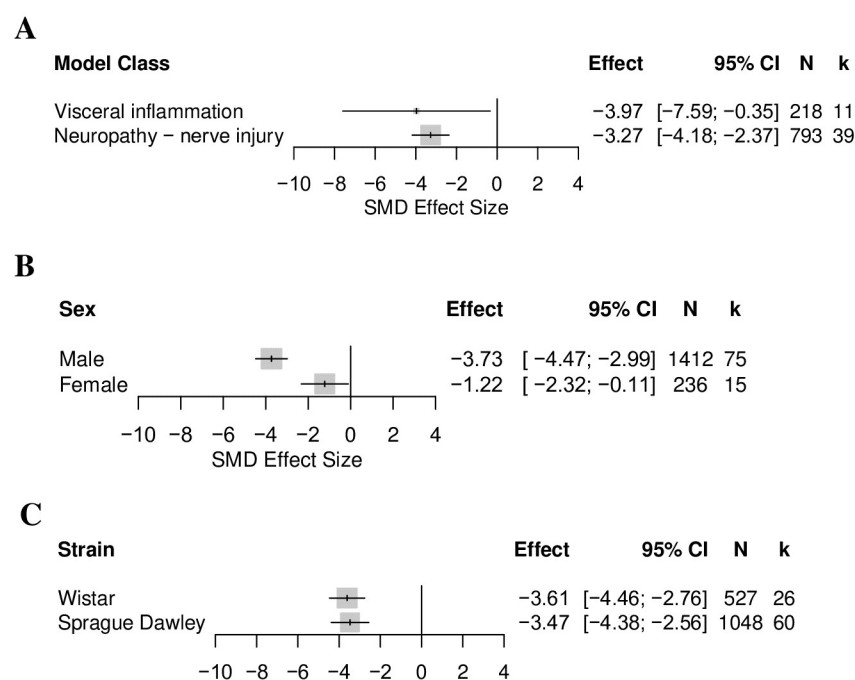

**Fig 6. Forest plots of the thigmotactic outcome in rats modelled with models associated with persistent pain: (A) model class, (B) sex, and (C) rat strain.** The size of the square represents the weight. CI, confidence interval; k, number of cohort-level comparisons; N, number of animals.

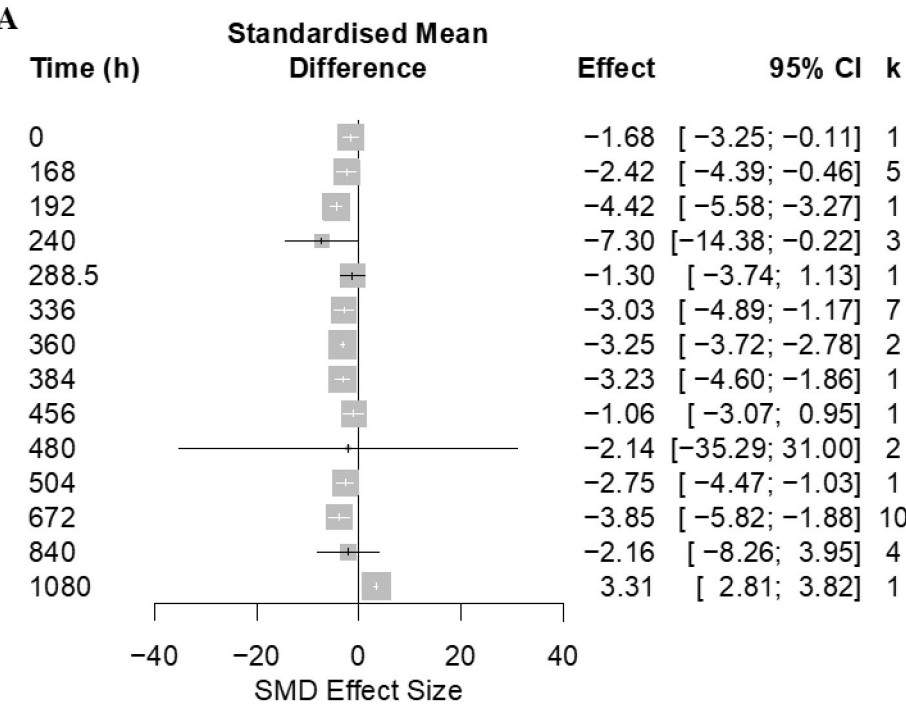

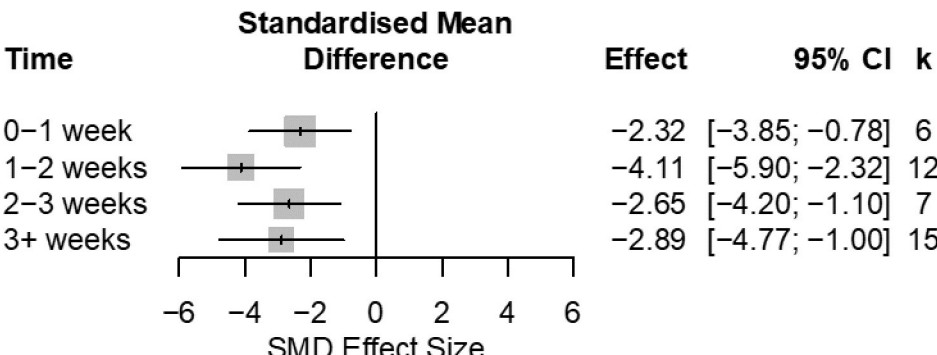

**Fig 7. Forest plots investigating impact of time since model induction on rat thigmotaxis.** Forty-one cohort-level comparisons assessed the impact of nerve injury models at (A) different time points (number of hours from model induction to the first OFT assessment) to sham control on rat thigmotaxis. (B) Time block analysis. For each comparison, an effect size was calculated using the Hedge's *g* SMD method. Effect sizes were pooled using the random effects model. The size of the square represents the weight, which reflects the contribution of each comparison with the pooled effect estimate. CI, confidence interval; k, number of cohort-level comparisons.

We could not perform a stratified meta-analysis to ascertain the effect of using different thigmotactic outcome metrics because of the predominance of "time spent in the centre" and little use of other metrics.

**2.3.2. Dataset 2: Effects of analgesic drug interventions on thigmotaxis outcomes in rat injury and disease models associated with persistent pain.** A total of 44 studies, containing 90 cohort-level comparisons, 1232 rats, and a sample size range from 8 to 48 with a median of

12 per group, assessed the effects of drug treatments from 35 drug classes on thigmotaxis within 14 types of models associated with persistent pain. Gabapentinoid was the most reported drug class (11%, k = 10). Male rats were used in 99% (k = 89) of experiments and female rats were used in 1% (k = 1). Eighty-eight percent (k = 79) of experiments assessed thigmotaxis by measuring time spent in the centre, 9% (k = 8) measured entries to the centre and 3% (k = 3) measured distance travelled in the centre. Drug treatments significantly reduced thigmotaxis in rat injury and disease models associated with persistent pain (SMD = 2.18 [95% CI 1.70 to 2.67]). Heterogeneity was high (Q = 1810.12, df = 89, $p$ < 0.0001, $I^2$ = 94%) (Fig 8).

*A positive correlation between time in the centre and total distance travelled in rat analgesic drug treatment experiments*. A total of 45 cohort-level comparisons assessed both time in the centre and total distance travelled in analgesic drug treatment experiments. Overall, total distance travelled was increased in drug experiments (SMD = 0.96 [95%CI 0.40 to 1.52]) (Fig 9). Total distance travelled was reduced the most in rats modelled with post-surgical pain that were given NMDA receptor antagonist (SMD = -2.51 [95%CI -3.53 to -1.49], k = 1), and was increased the most in rats modelled with nerve injury induced neuropathy that were given β-adrenoceptor antagonist (SMD = 4.28 [95%CI 2.62 to 5.94], k = 2). It should be noted that the prevalence of reporting for each drug class in each model class is low and may hinder the generalisability of such findings.

There was a moderate positive correlation between time in the centre and distance travelled (Coefficient = 0.42, df = 43, $p$ = 0.004) (Fig 10). To assess the effect of modelling, we could only analyse the correlation in rats modelled with nerve injury induced neuropathy (k = 18), and there was no correlation (Coefficient = 0.35, df = 16, $p$ = 0.16) (Fig 11). Insufficient cohort-level comparisons prevented further correlation tests of drug class and time in the centre and total distance travelled within the same model class.

*Unable to assess effects of animal model and animal characteristics on the thigmotactic outcome in rat analgesic drug treatment experiments*. We could not ascertain the effects of using different model classes, drug classes, sexes, and type of thigmotactic metrics on thigmotactic outcomes because only one condition from each variable has enough cohort-level comparisons. The full dataset can be accessed on Open Science Framework (OSF) (https://osf.io/xqgsc). However, the effect of strain could be assessed. Of the six strains reported, stratified meta-analysis of Sprague-Dawley (k = 43) and Wistar (k = 40) rats showed strain that did not account for a significant proportion of heterogeneity (Q = 0.00, df = 1, $p$ = 0.98) (Fig 12).

**2.3.3. Dataset 3: Effects of modelling persistent pain on thigmotaxis outcomes in mouse experiments using sham controls.** A total of 53 studies, containing 116 cohort-level comparisons, 2012 mice, and a sample size range from 7 to 63 with a median of 13.5 animals per group, assessed the effects of 16 types of disease models associated with persistent pain on thigmotaxis. Seventy-two percent (k = 83) of experiments assessed thigmotaxis by measuring time spent in the centre, 23% (k = 27) measured time spent in the periphery and 5% (k = 6) measured distance travelled in the centre. Thigmotaxis was increased by injury and disease models associated with persistent pain compared to sham controls in mice (SMD = -1.23 [95%CI -1.80 to -0.67]). Heterogeneity was high (Q = 4316.72, *df* = 115, $p$ < 0.0001, $I^2$ = 97%) (Fig 13).

*No correlation between time in the centre, time in the periphery and total distance travelled in mouse modelling experiments compared to sham controls*. A total of 49 cohort-level comparisons assessed both time in the centre and total distance travelled in animal modelling experiments. Overall, total distance travelled was reduced by injury and disease models associated with persistent pain (SMD = -1.50 [95%CI -2.93 to -0.07]) (Fig 14). Total distance travelled was reduced the most in mice with P2Y12 deficiency (SMD = -5.95 [95%CI -6.83 to -5.07], k = 1), and was increased the most in mice modelled with traumatic brain injury (SMD = 2.30 [95%CI 0.68 to 3.92], k = 1).

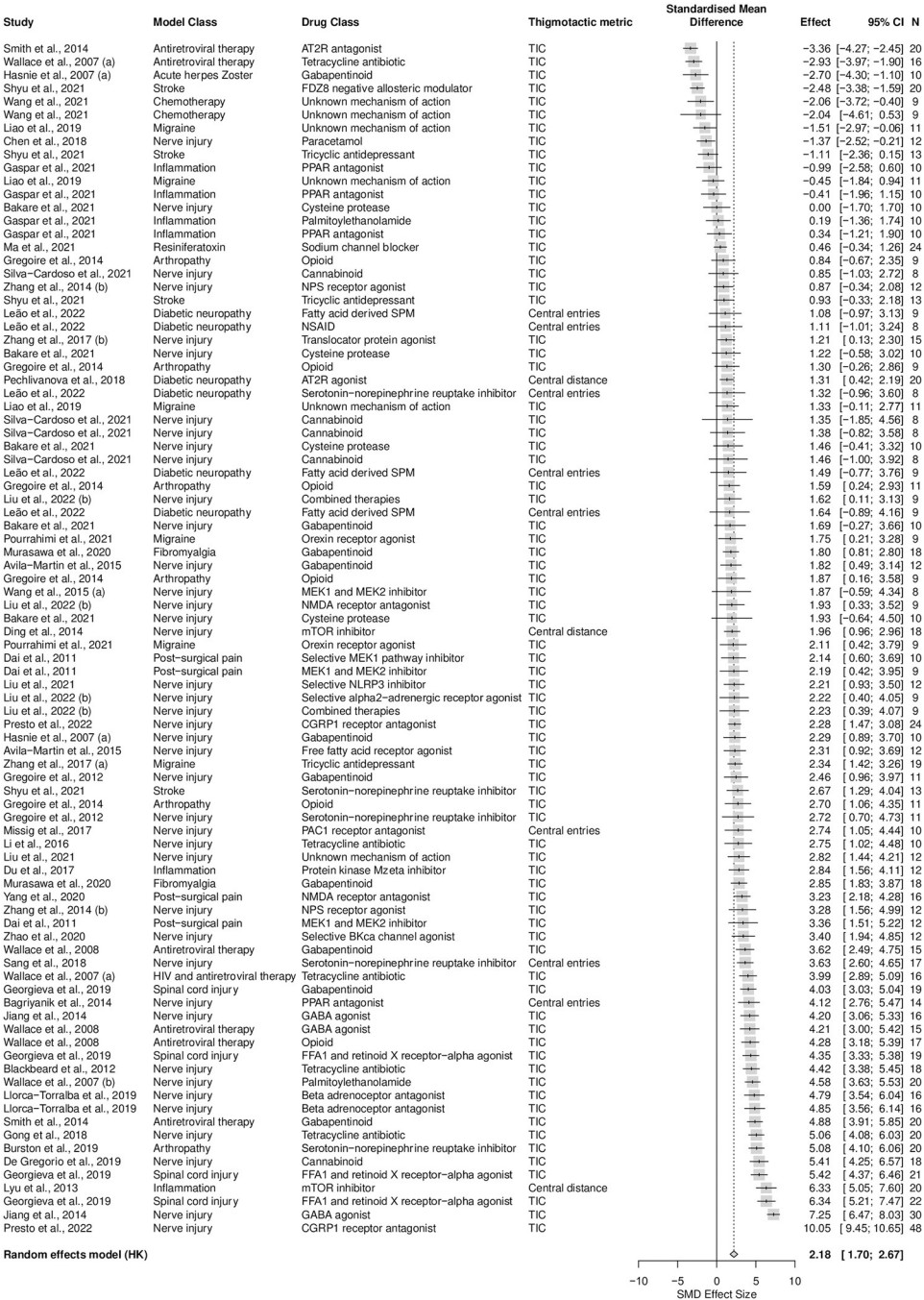

**Fig 8. Forest plot of drug treatment effects on rat thigmotaxis.** A summary forest plot of the 90 cohort-level comparisons which assessed the drug treatment effects on rat thigmotaxis. For each comparison, an effect size was calculated using the Hedges' g SMD method. Effect sizes were pooled using the random effects model. The restricted maximum-likelihood method was used to estimate heterogeneity. The size of the square represents the weight, which reflects the contribution of each comparison with the pooled effect estimate. TIC, time spent in the centre. CI, confidence interval; N, number of animals.

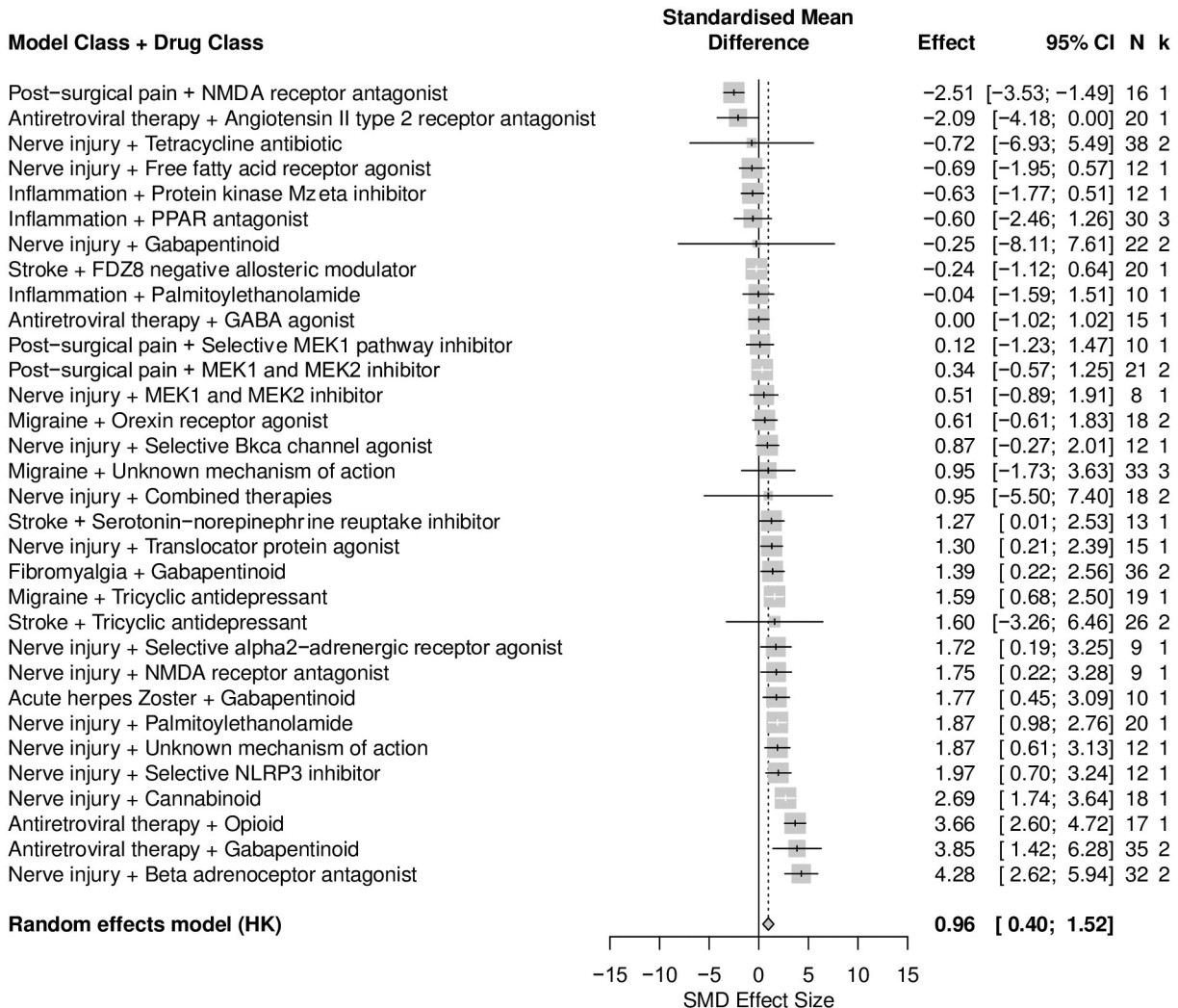

**Fig 9. Forest plot of impact of modelling and drug treatment on total distance travelled in rats.** Forty-five cohort-level comparisons assessed the impact of modelling and drug treatment on total distance travelled in rats. The overall effect size is 0.96 [95% CI 0.40 to 1.52]. The size of the square represents the weight, which reflects the contribution of each comparison with the pooled effect estimate. CI, confidence interval; k, number of cohort-level comparisons; N, number of animals.

There was no correlation between time in the centre and total distance travelled (Coefficient = 0.26, df = 47, *p* = 0.07). (Fig 15). In mice modelled with nerve injury neuropathy (k = 24) there wasn't a correlation (Coefficient = -0.02, df = 22, *p* = 0.94) (Fig 16).

A total of 26 cohort-level comparisons assessed both time in the periphery and total distance travelled in mouse modelling experiments, and there was no correlation (Coefficient = 0.30, df = 24, *p* = 0.14) (Fig 17). We also assessed the correlation between time in the periphery and total distance travelled in mice modelled with nerve injury induced neuropathy (k = 13) (Fig 18A) and inflammation (k = 11) (Fig 18B), and there wasn't a correlation (Coefficient = 0.13, df = 11, *p* = 0.66; Coefficient = 0.33, df = 9, *p* = 0.32, respectively).

*Effects of animal model and animal characteristics on the thigmotactic outcome in mouse modelling experiments.* Nerve injury induced neuropathy (k = 50), inflammation (k = 19) and spinal cord injury (k = 11) were the three model classes with enough comparisons for a

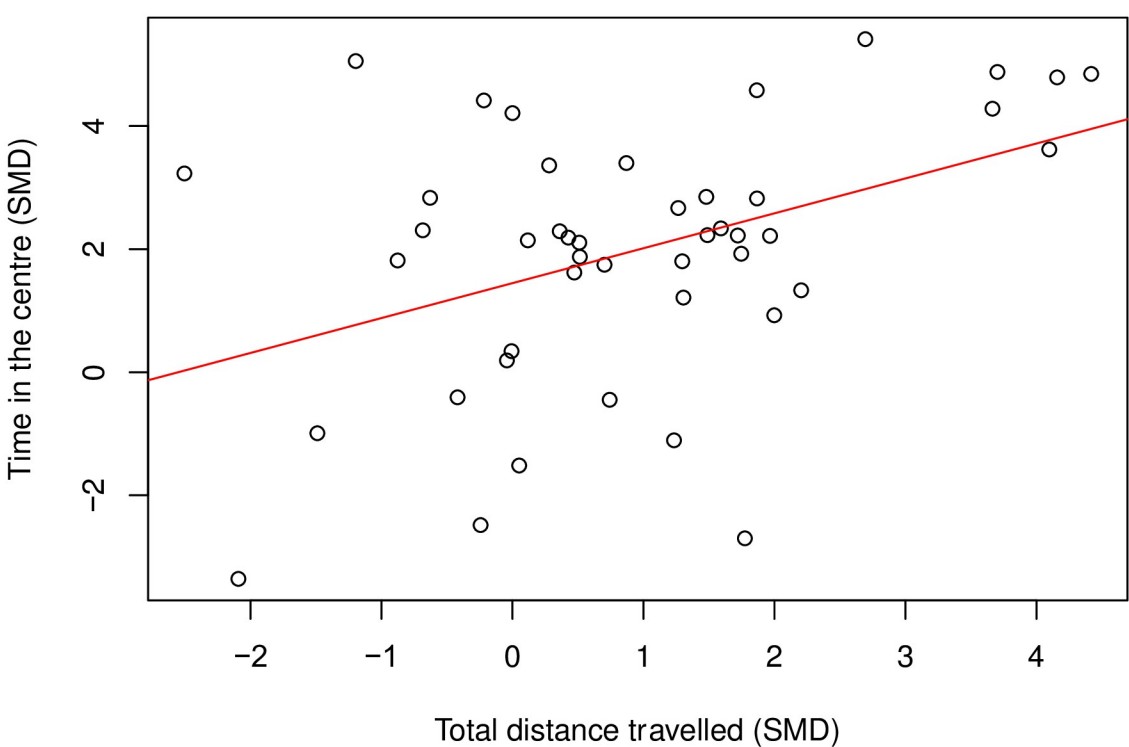

**Fig 10. A Pearson's Correlation test between time in the centre and total distance travelled in rat drug treatment experiments.** There is a moderate positive correlation (Coefficient = 0.42, df = 43, $p$ = 0.004, k = 45). A line of best fit (in red) was drawn. SMD, standardised mean difference.

stratified analysis. The model class did not account for a significant proportion of heterogeneity (Q = 0.62, df = 2, $p$ = 0.73) (Fig 19A). Only mice modelled with nerve injury induced neuropathy significantly increased thigmotaxis (SMD = 01.27 [95%CI -2.23 to -0.30]).

Male mice were used in 71% (k = 82) of experiments, female mice were used in 25% (k = 29), 3% (k = 4) used mixed sexes, and 1% (k = 1) did not report the sex of the mice used. The experiment in which sex was not reported, and the 4 experiments which reported mixed sexes were not included in the stratified meta-analysis. Sex did not account for a significant proportion of heterogeneity (Q = 0.04, df = 1, $p$ = 0.84) (Fig 19B). Thigmotaxis was more significantly increased in male mice.

Twelve mouse strains were reported. C57BL/6 was the most reported strain (78%, k = 90). It was not possible to ascertain the effect of strain on the thigmotactic outcome because of too few cohort-level comparisons for a stratified meta-analysis.

Similarly, we could not perform a stratified analysis to ascertain the effect of using different thigmotactic because only time spent in the centre and time spent in the periphery have enough cohort-level comparisons, and they are obverse of each other.

Open field tests were conducted between 24 h and 180 days after nerve injury (for which there were 46 comparisons that reported time since model induction). All time points accounted for a significant proportion of heterogeneity (Q = 634.39; d.f. = 19; p <0.0001) (Fig 20A). The same was observed in the time block analysis (Q = 16.61; d.f. = 3; p = 0.0009) and the 2–3 week time block resulted in the largest observed effect sizes (Fig 20B). However, these findings should be interpreted with caution given the low number of comparisons for each time period.

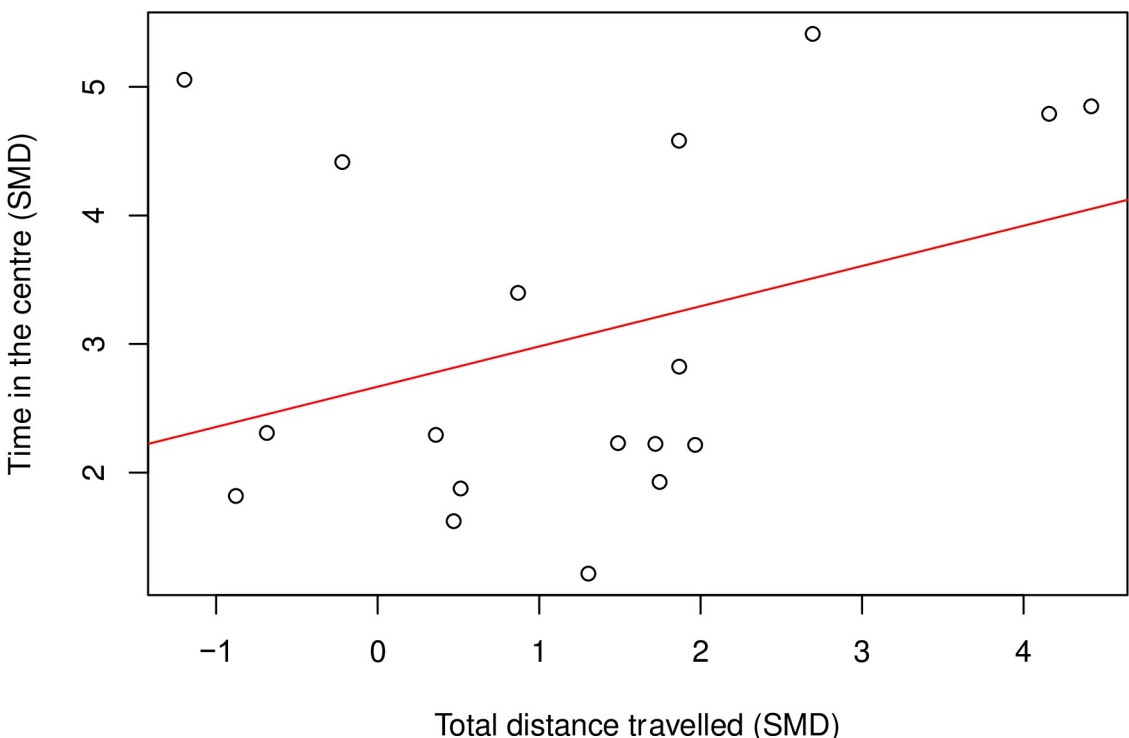

**Fig 11. A Pearson's Correlation test between time in the centre and total distance travelled in drug treatment experiments in rats modelled with nerve injury.** There is no correlation (Coefficient = 0.35, df = 16, $p$ = 0.16, k = 18). A line of best fit (in red) was drawn. SMD, standardised mean difference.

**2.3.4. Dataset 4: Effects of analgesic drug interventions on thigmotaxis outcomes in mouse models of persistent pain.** A total of 27 studies, containing 57 cohort-level comparisons, 823 rats, and a sample size range from 8 to 30 with a median of 14 per group, assessed the effects of drug treatments from 25 drug classes on thigmotaxis in 11 types of models associated with persistent pain. Hydrogen sulfide donor was the most reported drug class (12%, k = 7). Male mice were used in 81% (k = 46) of experiments, female mice were used in 14% (k = 8) and 5% (k = 3) of experiments used mixed sexes. There were 4 different mouse strains reported, and C57BL/6 was the most reported strain (86%, k = 49). Sixty-eight percent (k = 39)

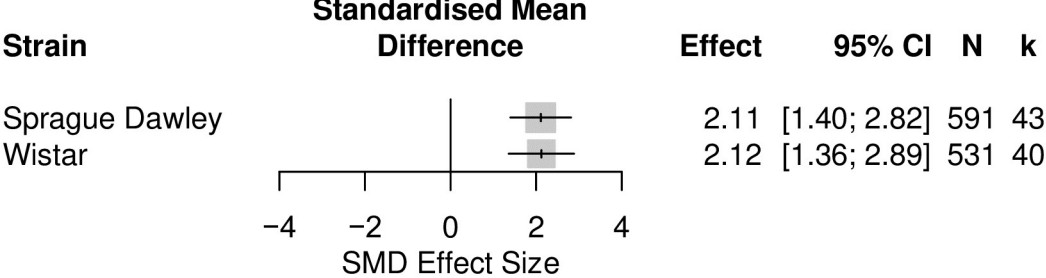

**Fig 12. A forest plot of the drug treatment effect on thigmotactic outcome in Wistar and Sprague-Dawley rats induced with models associated with persistent pain.** The size of the square represents the weight. CI, confidence interval; k, number of cohort-level comparisons; N, number of animals.

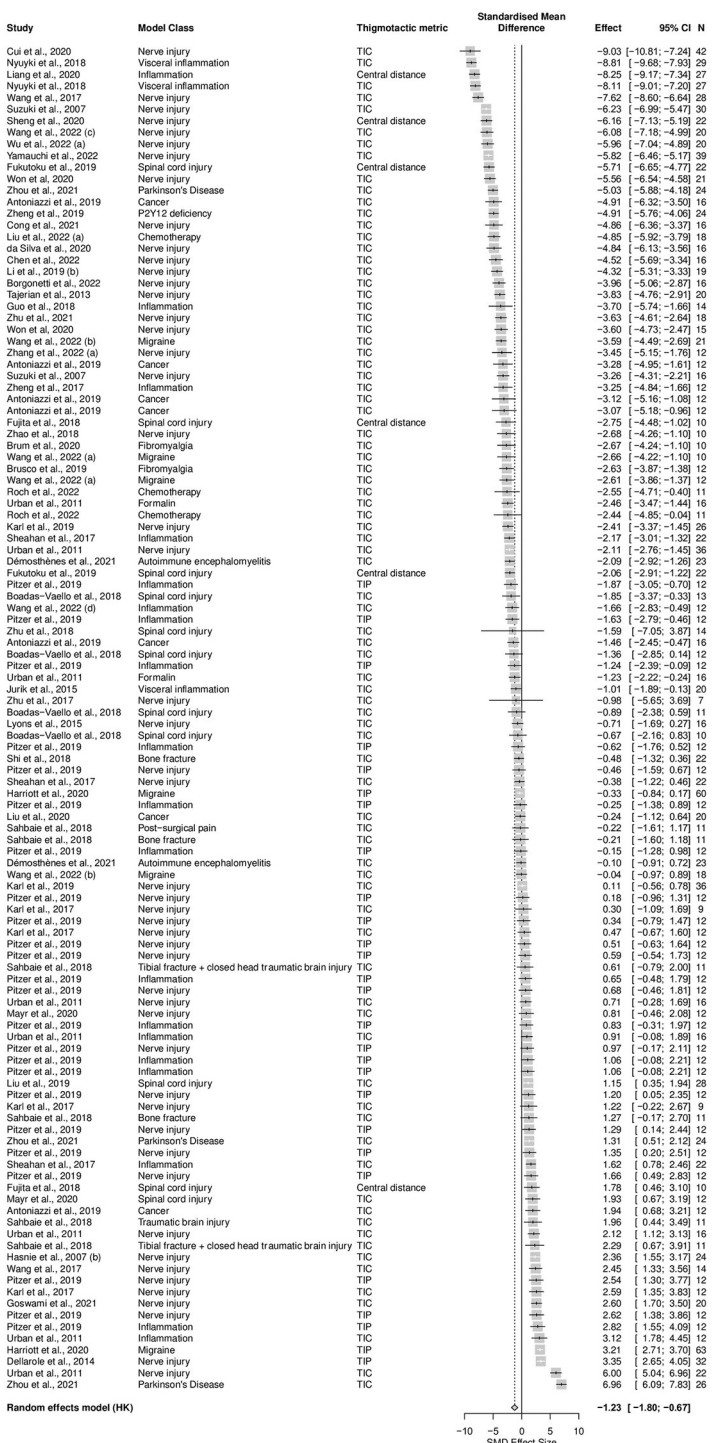

**Fig 13. Forest plot of impact of modelling on mouse thigmotaxis.** A summary forest plot of the 116 cohort-level comparisons which assessed the impact of modelling compared to sham control on mouse thigmotaxis. For each comparison, an effect size was calculated using the Hedges' *g* SMD method. Effect sizes were pooled using the random effects model. The restricted maximum-likelihood method was used to estimate heterogeneity. The overall effect size is -1.23 [95%CI -1.80 to -0.67]; Q = 4316.72, *df* = 115, *p* < 0.0001, $I^2$ = 97%. The size of the square represents the weight, which reflects the contribution of each comparison with the pooled effect estimate. TIC, time spent in the centre; TIP, time spent in the periphery. CI, confidence interval; N, number of animals.

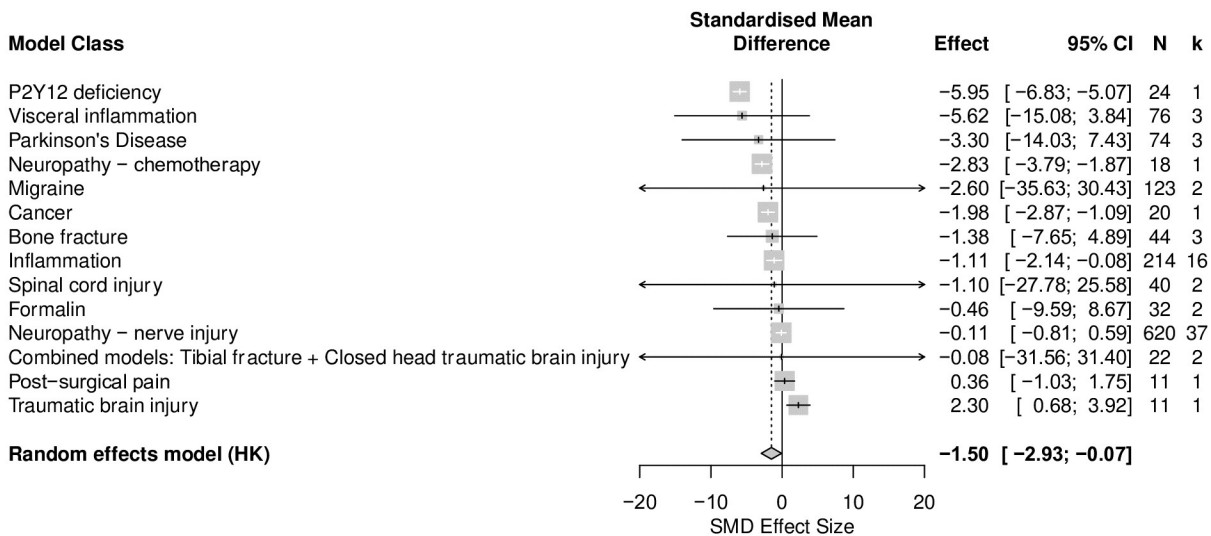

**Fig 14. Forest plot of impact of modelling on total distance travelled in mice.** Forty-nine cohort comparisons assessed the impact of modelling on total distance travelled in mice. The overall effect size is -1.50 [95% CI -2.93 to -0.07]. The size of the square represents the weight, which reflects the contribution of each comparison with the pooled effect estimate. CI, confidence interval; k, number of cohort-level comparisons; N, number of animals.

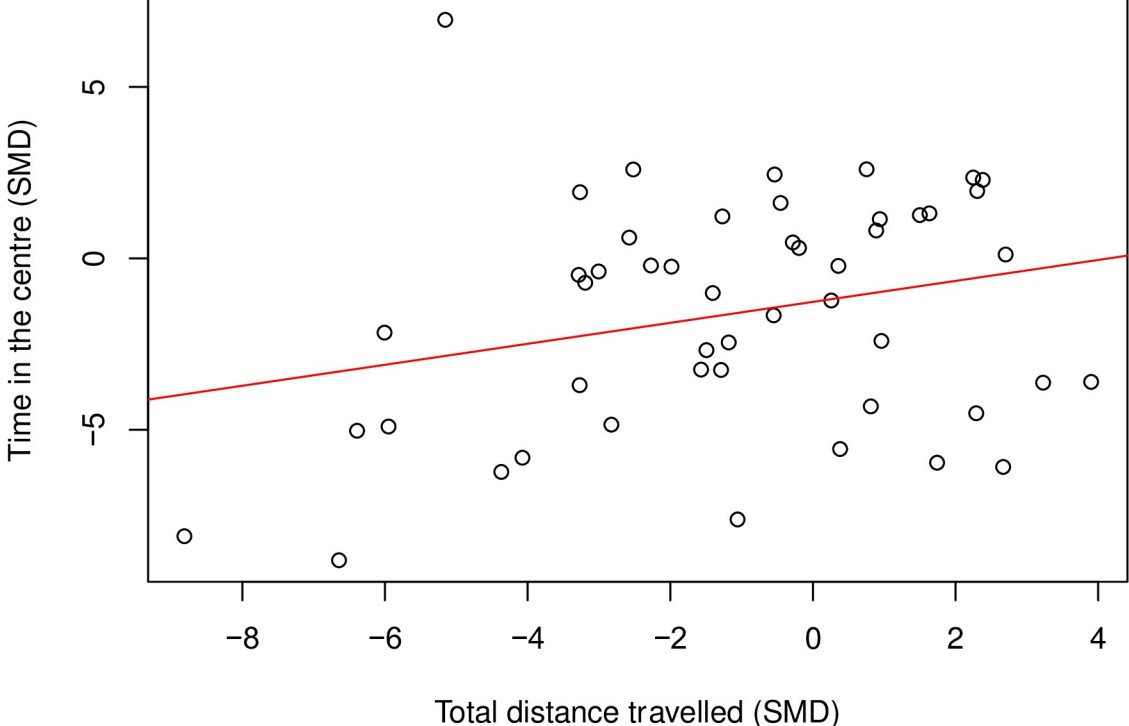

**Fig 15. A Pearson's Correlation test between time in the centre and total distance travelled in mouse modelling experiments.** No correlation: Coefficient = 0.26, df = 47, $p$ = 0.07, k = 49). A line of best fit (in red) was drawn. SMD, standardised mean difference.

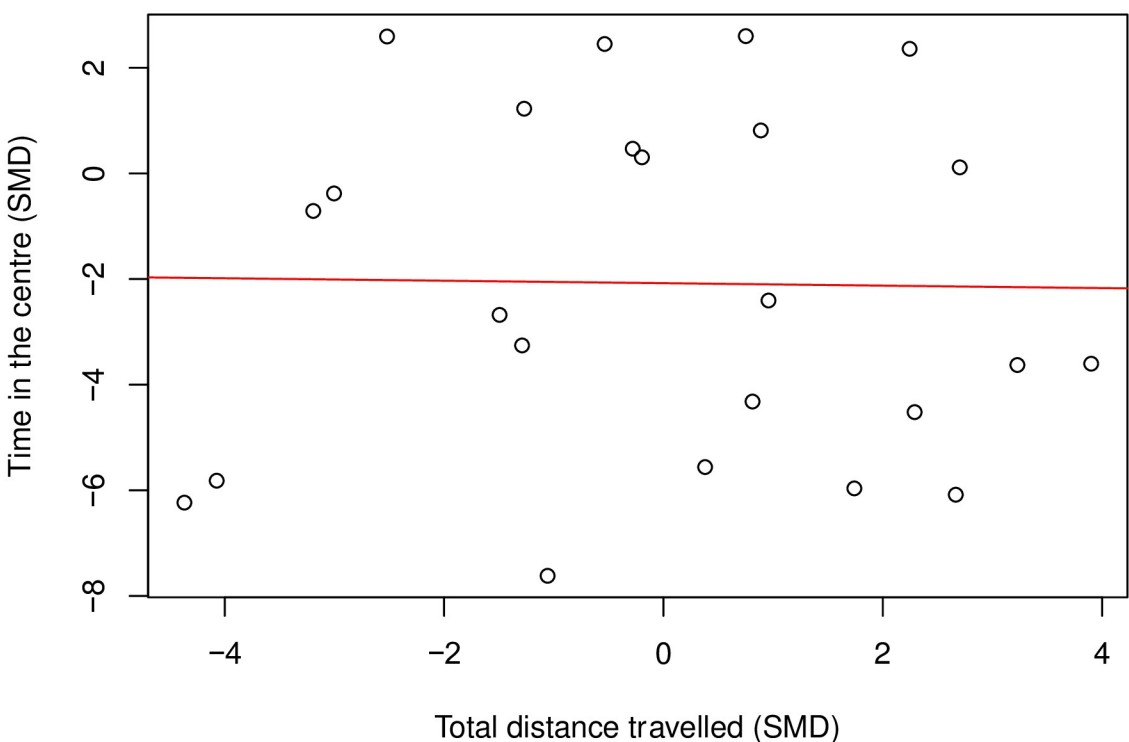

**Fig 16. A Pearson's Correlation test between time in the centre and total distance travelled in mice modelled with nerve injury induced neuropathy.** There is no correlation (Coefficient = -0.02, df = 22, $p$ = 0.94, k = 24). A line of best fit (in red) was drawn. SMD, standardised mean difference.

of experiments assessed thigmotaxis by measuring time spent in the centre, 23% (k = 13) measured distance travelled in the centre, 4% (k = 2) measured time spent in the periphery, 4% (k = 2) measured centre perimeter ratio and 1% (k = 1) measured periphery/centre ratio.

Increased thigmotaxis caused by injury and disease models associated with persistent pain was reduced by analgesic drug treatments in mice (SMD = 2.18 [95%CI 1.70 to 2.67]). Heterogeneity was high (Q = 1810.12, df = 89, $p$ < 0.0001, $I^2$ = 94%) (Fig 21).

*No correlation between time in the centre or distance travelled in the centre and total distance travelled in analgesic drug treatment experiments.* Nineteen cohort-level comparisons assessed time in the centre and total distance travelled, and 13 cohort level comparisons assessed distance travelled in the centre and total distance travelled in analgesic drug treatment experiments. Overall, total distance travelled was increased in drug experiments (SMD = 1.33 [95%CI 0.42 to 2.24]) (Fig 22). The total distance travelled was reduced the most in mice modelled with nerve injury that were given analgesics with unknown mechanism of action (SMD = -2.81 [95%CI -4.08 to -1.54], k = 1), and was increased the most in mice modelled with inflammation that were given acetylcholinesterase inhibitors (SMD = 4.13 [95%CI 3.79 to 4.47], k = 3).

There was no correlation between time in the centre and total distance travelled (Coefficient = 0.26, df = 17, $p$ = 0.29) (Fig 23A) nor between distance travelled in the centre and total distance travelled (Coefficient = 0.006, df = 11, $p$ = 0.99, k = 13) (Fig 23B). There is an insufficient number of cohort-level comparisons, so we could not assess the correlation between time in the centre and total distance travelled within the same model and drug class.

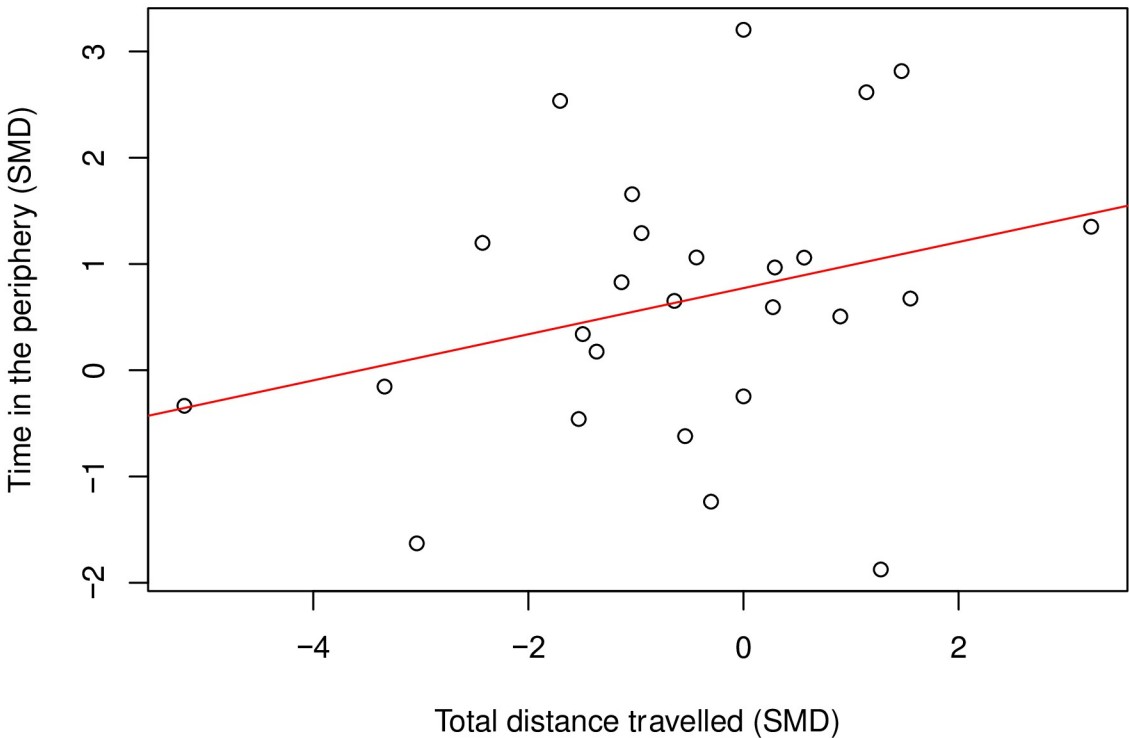

**Fig 17. A Pearson's correlation test between time in the periphery and total distance travelled in mouse modelling experiments.**
No correlation: Coefficient = 0.30, df = 24, *p* = 0.14, k = 26). A line of best fit (in red) was drawn. SMD, standardised mean difference.

*Effects of animal model and animal characteristics on the thigmotactic outcome in mouse analgesic drug treatment experiment.* Nerve injury induced neuropathy (k = 13), inflammation (k = 12) and spinal cord injury (k = 11) were the model classes with enough comparisons for stratified analyses. The model type did not account for a significant proportion of heterogeneity (Q = 4.19, df = 2, *p* = 0.12) (Fig 22A). Drug treatments significantly reduced thigmotaxis in mice modelled with inflammation and nerve injury induced neuropathy. We could not ascertain the effects of sex, strain, and drug classes because there were not enough cohort-level comparisons (k<10). The type of thigmotactic metric; time in the centre (k = 39) and distance travelled in the centre (k = 13) did not account for a significant proportion of heterogeneity (Q = 0.60, df = 1, *p* = 0.44) (Fig 24B).

**2.3.5. Effects of the experimental conditions and apparatus on thigmotactic outcome.**
We also assessed the effects of experimental design variables related to the experimental conditions and the OF apparatus however, the analysis was limited by the low levels of reporting. Where it was possible to make comparisons, the difference for most variables was not statistically significant and the observed variability could not be explained (Table S3.1 in S3 File).

**2.3.6. Correlation of thigmotaxis and stimulus-evoked limb withdrawal.** To discern the value of thigmotaxis in pain research, we assessed the correlation between type of thigmotactic metric and type of stimulus-evoked limb withdrawal. The number of studies that investigated both thigmotaxis and stimulus-evoked limb withdrawal for each dataset are summarised in (Table S3.2 in S3 File).. To summarise our findings, there was not a correlation between time in the centre and mechanical included behavioural outcomes for rat modelling experiments but there was a positive correlation in rat analgesic drug treatment experiments and mouse modelling experiments. There is an insufficient number of comparisons to assess the

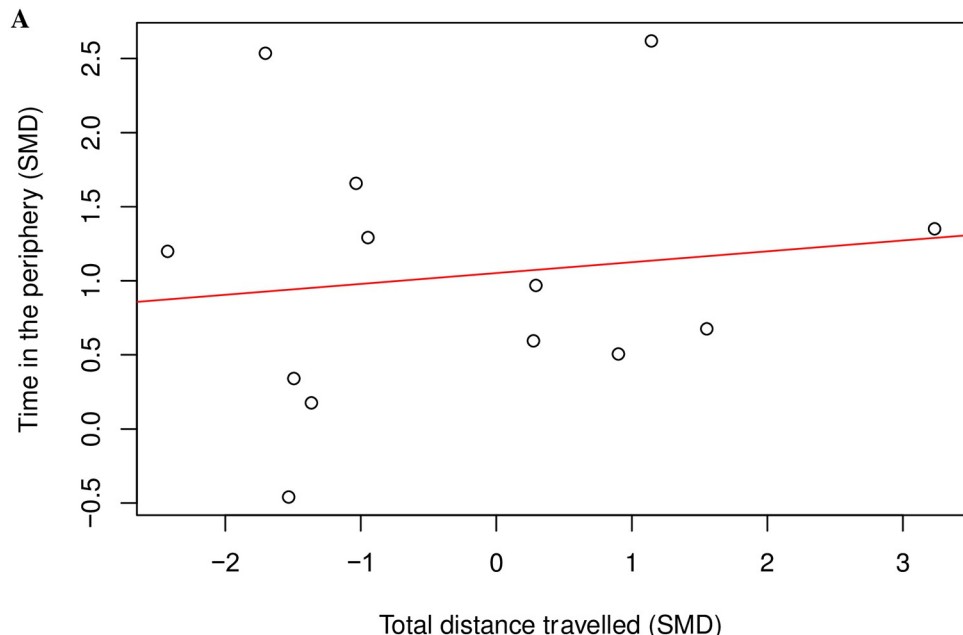

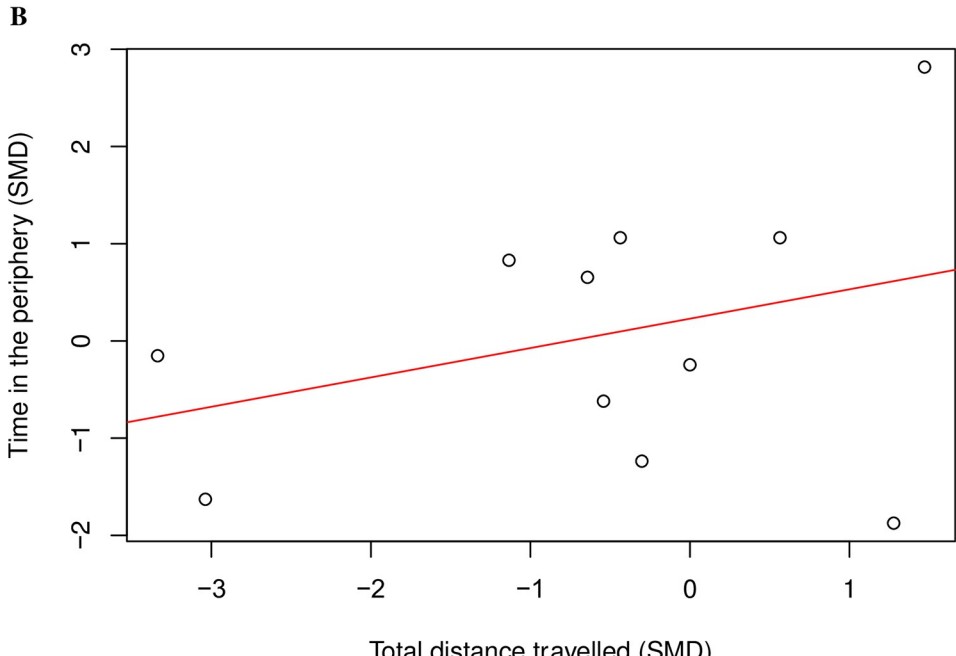

**Fig 18. A Pearson's correlation test between time in the periphery and total distance travelled.** (A) in mice modelled with nerve injury induced neuropathy (no correlation; Coefficient = 0.13, df = 11, $p$ = 0.66, k = 13); and (B) in mice modelled with inflammation (no correlation; Coefficient = 0.33, df = 9, $p$ = 0.32, k = 11). A line of best fit (in red) was drawn. SMD, standardised mean difference.

correlation between the type of thigmotactic metric and the type of stimulus-evoked limb withdrawal in mouse analgesic drug treatment experiments.

When possible, we also assessed the correlation between thigmotaxis and the type of the stimulus-evoked limb withdrawal in rodent species that received the same treatments

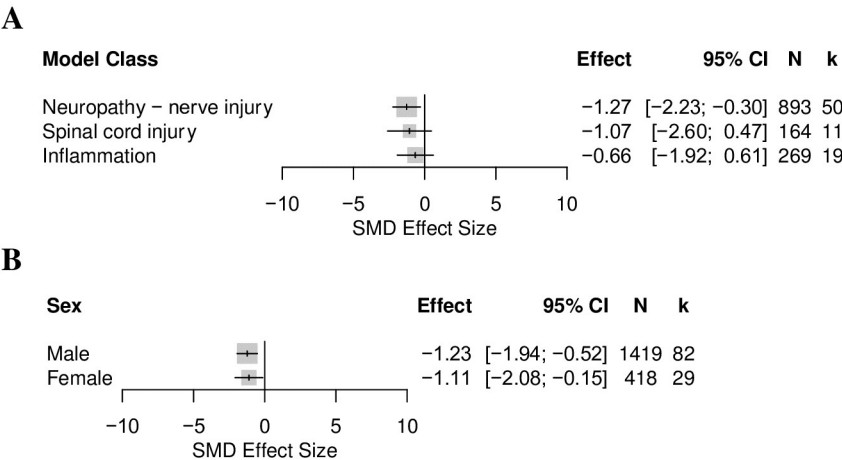

**Fig 19. Forest plots of the thigmotactic outcome in mice modelled with models associated with persistent pain: (A) model class and (B) sex.** The size of the square represents the weight. CI, confidence interval; k, number of cohort-level comparisons; N, number of animals.

(Table S3.3 in S3 File). We identified strong positive correlations between time in the centre and mechanical induced behavioural outcomes in rats modelled with nerve injury induced and analgesic drug treatment. No correlation was found between time in the centre and mechanical induced behavioural outcomes in mice modelled with nerve injury and we are unable to ascertain the effect of drug class on this correlation due to having insufficient number of comparisons (S3 File).

**2.3.7. Risk of bias.** The overall risk of bias of the 181 studies is unclear. The reporting of random group allocation (49%, 88 studies) and blinding of outcome assessment (47%, 85 studies) were low. The reporting of other methodological quality criteria was very low: 8% (16 studies) reported allocation concealment, 12% (21 studies) reported sample size calculation, 19% (35 studies) reported pre-defined animal inclusion criteria, and 15% (27 studies) reported animal inclusion (Fig 25A). This contrasts with the high reporting of conflict of interest (85%, 153 studies) and compliance with animal welfare regulations (99%, 179 studies). The methods used to mitigate bias were rarely reported hence an unclear risk of bias (Fig 25B). A traffic light plot presenting the risk of bias score for each report is available in S4 File.

**2.3.8. Impact of methodological quality criteria on thigmotactic effect sizes.** Thigmotactic effect sizes of rats and mice were combined to assess the impact of each criterion. In animal modelling experiments, only the reporting of pre-defined animal inclusion criteria accounted for a significant proportion of the observed heterogeneity (Fig 26). Larger effect sizes were observed in experiments that reported inclusion criteria (SMD = -3.34 vs -1.92, Q = 4.15, df = 1, $p$ = 0.04). It is noteworthy that the prevalence of reporting inclusion criteria was relatively low (k = 35 reported vs k = 172 not reported) so may limit our ability to accurately determine its influence on thigmotaxis.

In drug treatment experiments, reporting of the six methodological quality criteria did not account for a significant proportion of the observed heterogeneity (Fig 27).

**2.3.9. Reporting quality.** Out of 181 included studies, 13 studies were published before the introduction of the ARRIVE guidelines in 2010. Twenty-five studies (14%) stated reporting in accordance with the ARRIVE guidelines, but only 3 studies provided a checklist. Of these, 2 studies reported sufficient details on the methods used to mitigate bias and were scored a low risk of bias, however one of the studies did not report allocation concealment, sample size

**A**

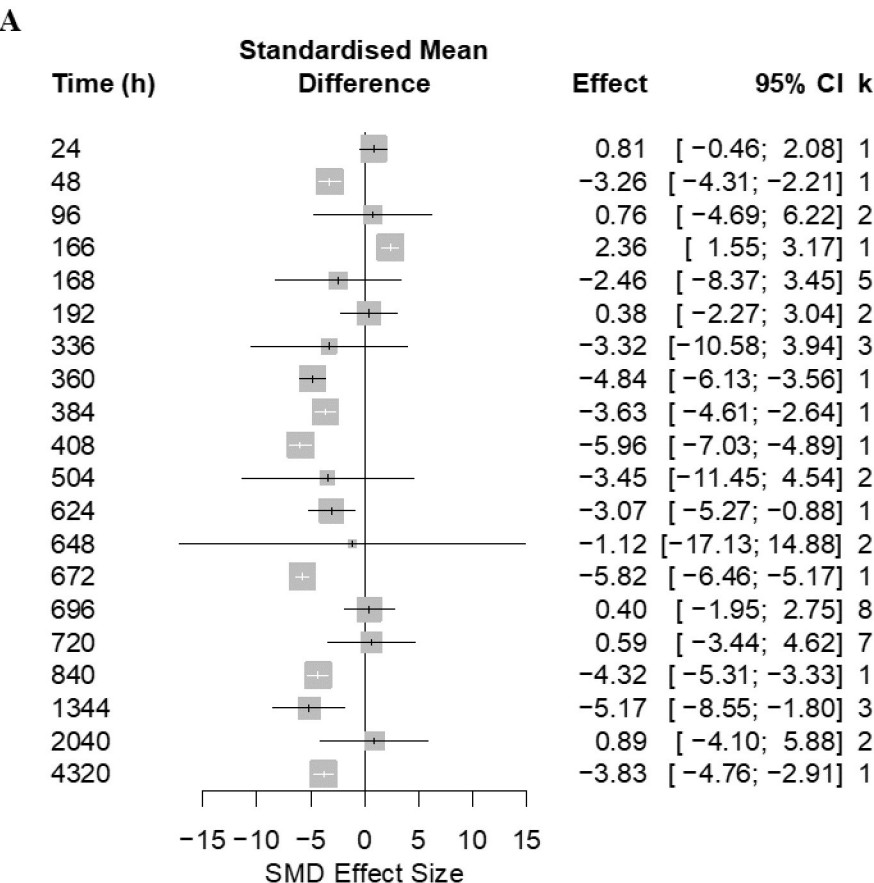

**B**

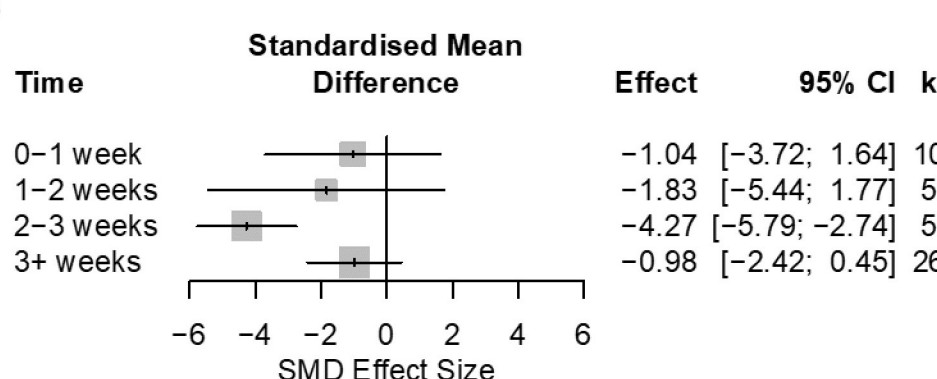

**Fig 20. Forest plot of impact of nerve injury models and time since induction on mouse thigmotaxis.** A summary forest plot of the 46 cohort-level comparisons which assessed the impact of nerve injury models compared at different time points (number of hours from model induction to the first OFT assessment) to sham control on mouse thigmotaxis. For each comparison, an effect size was calculated using the Hedge's *g* SMD method. Effect sizes were pooled using the random effects model. The size of the square represents the weight, which reflects the contribution of each comparison with the pooled effect estimate. CI, confidence interval; k, number of cohort-level comparisons.

calculation, pre-defined animal inclusion criteria and animal exclusions despite providing the checklist. Furthermore, this study did not sufficiently report the method details for randomisation and blinding, hence it was scored an unclear risk of bias. For the 22 studies that stated reporting in accordance with the ARRIVE guidelines but did not provide a checklist, the

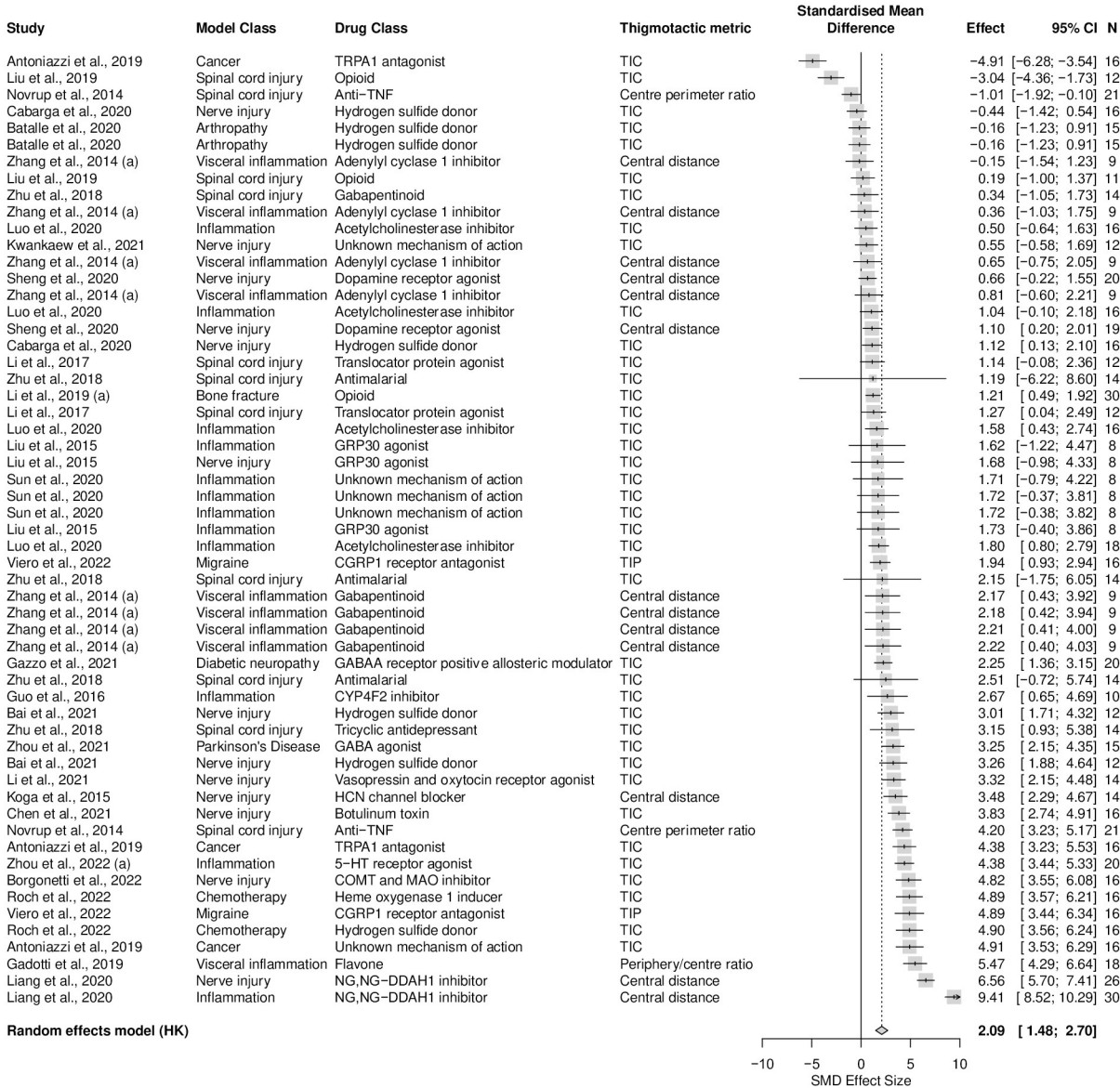

**Fig 21. Forest plot of drug treatment effects on mouse thigmotaxis.** A summary forest plot of the 57 cohort-level comparisons which assessed the drug treatment effects on mouse thigmotaxis. For each comparison, an effect size was calculated using the Hedges' *g* SMD method. Effect sizes were pooled using the random effects model. The restricted maximum-likelihood method was used to estimate heterogeneity. The size of the square represents the weight, which reflects the contribution of each comparison with the pooled effect estimate. CI, confidence interval; N, number of animals.

reporting of methodological quality criteria was low to moderate, and the details of methods were mostly insufficiently reported (Table 3).

**2.3.10. Other sources of biases.** We conducted analyses to assess other sources of biases. The overall effect size when combining animal modelling thigmotactic data of rats and mice (k = 207) is -2.16 [95%CI -2.61 to -1.71]. Egger's regression test was significant ($p < 0.0001$), suggesting the presence of funnel plot asymmetry (Fig 28A). The overall effect size of combined rats and mice drug treatment thigmotactic data (k = 147) is 2.12 [95%CI 1.75 to 2.49]. Egger's regression test was significant ($p < 0.0001$), suggesting the presence of funnel plot

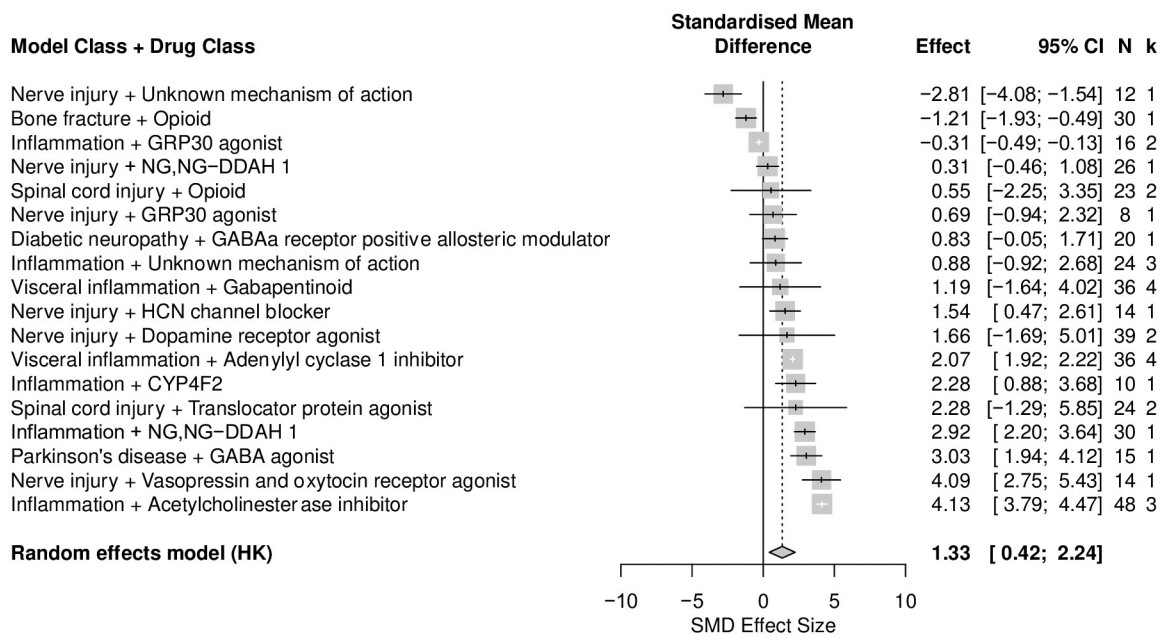

**Fig 22. Forest plot of impact of modelling and drug treatments on total distance travelled in mice.** Thirty-two cohort-level comparisons assessed the impact. The overall effect size is 1.33 [95%CI 0.42 to 2.24]. The size of the square represents the weight, which reflects the contribution of each comparison with the pooled effect estimate. CI, confidence interval; k, number of cohort-level comparisons; N, number of animals.

asymmetry (Fig 28B). Trim-and-fill analysis, however, did not impute theoretically missing experiments.

## 3. Discussion

We have presented a systematic and unbiased assessment of the published literature regarding the sensitivity of thigmotactic behaviour in laboratory rodents to a range of injury and disease models associated with persistent pain and the effects of analgesic drug interventions.

This systematic review identified 181 studies. Of which 165 studies were included in the meta-analysis, comprising the effects of 66 persistent pain associated injury or disease models and 80 potential analgesic interventions on thigmotactic behaviour in 5998 rodents. We report that, overall, thigmotaxis was increased in animal models associated with persistent pain and this increase was attenuated by analgesic drug treatments in both rat and mouse experiments. Stratified subgroup analyses and assessment of the impact of experimental design characteristics was limited due to the paucity of data and low levels of reporting on experimental design characteristics. In addition, there is overall a high level of unexplained heterogeneity and unclear risk of bias.

### 3.1. Limited ability to assess and identify sources of heterogeneity that influence thigmotaxis

There is a significant amount of observed heterogeneity that could not be explained by subgroup analyses or meta-regressions., The predominance of nerve injury and inflammation models, male animals, single strains, and breadth of novel compounds has limited our ability to identify variables that influence thigmotaxis to guide future research. Given the broad range

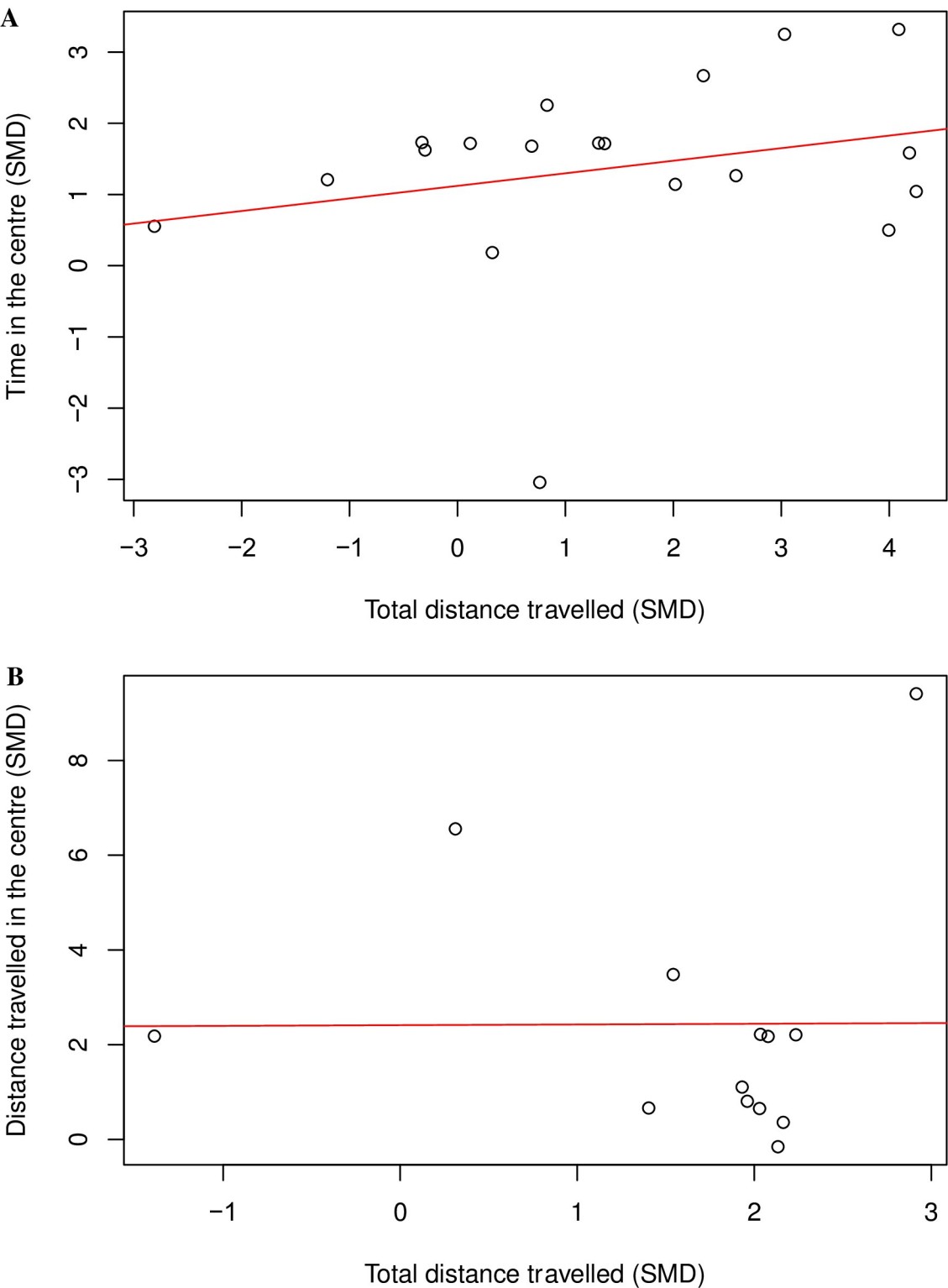

**Fig 23. A Pearson's Correlation test between (A) time in the centre and total distance travelled and (B) distance travelled in the centre and total distance travelled in mouse drug treatment experiments.** There is no correlation; Coefficient = 0.26, df = 17, $p$ = 0.29, k = 19 and Coefficient = 0.006, df = 11, $p$ = 0.99, k = 13 respectively. A line of best fit (in red) was drawn. SMD, standardised mean difference.

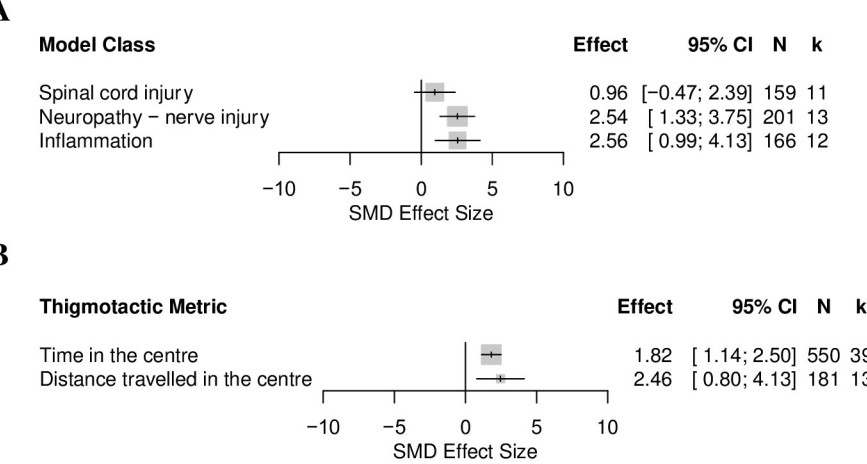

**Fig 24. Forest plots of the drug treatment effect on thigmotactic outcome in mice modelled with models associated with persistent pain: (A) model class and (B) type of thigmotactic metric.** The size of the square represents the weight. CI, confidence interval; k, number of cohort-level comparisons; N, number of animals.

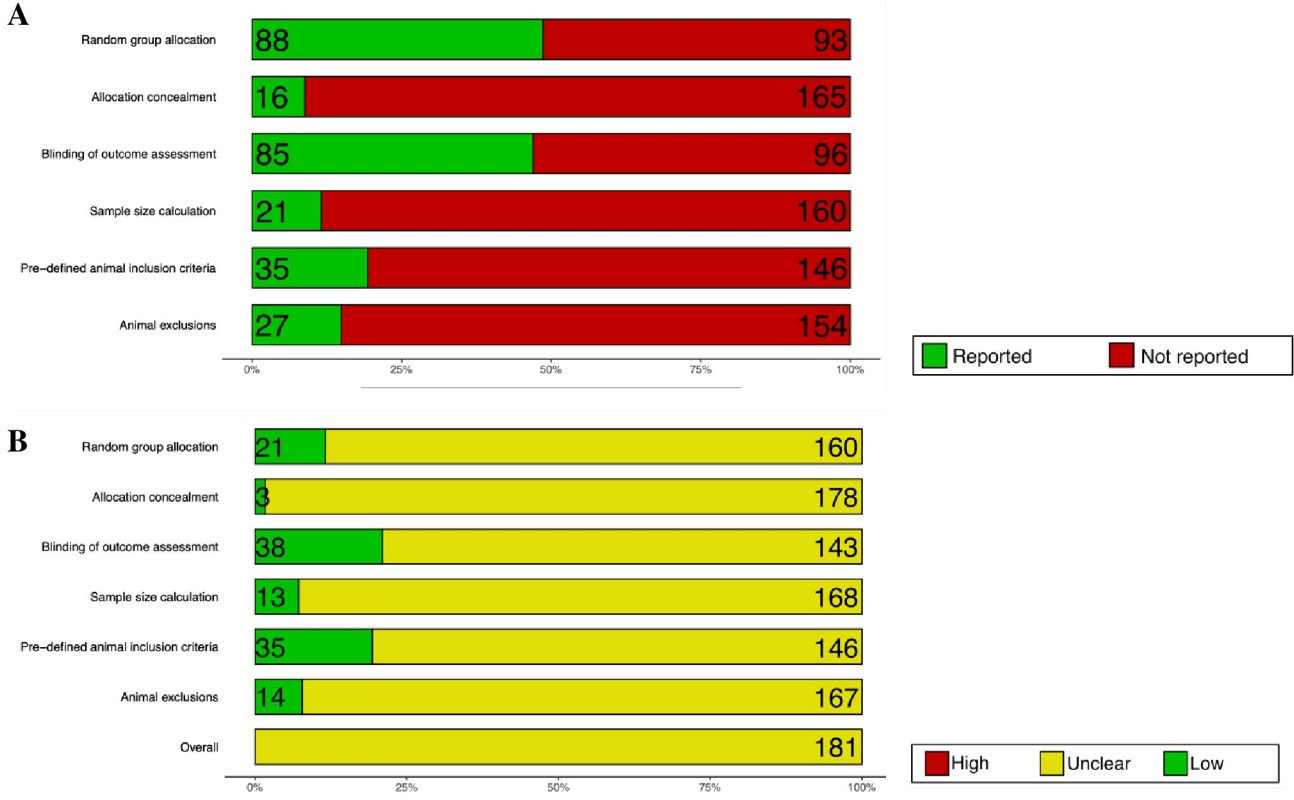

**Fig 25. Risk of bias.** Summary plots showing the percentage of the 181 studies that (A) reported the methodological quality criteria and (B) the corresponding risk of bias score given for each methodological quality criterion. Numbers shown within the bar plots indicate the number of studies. Reporting of a statement regarding potential conflict of interests and compliance with animal welfare regulations were extracted, but they were not part of the overall risk of bias.

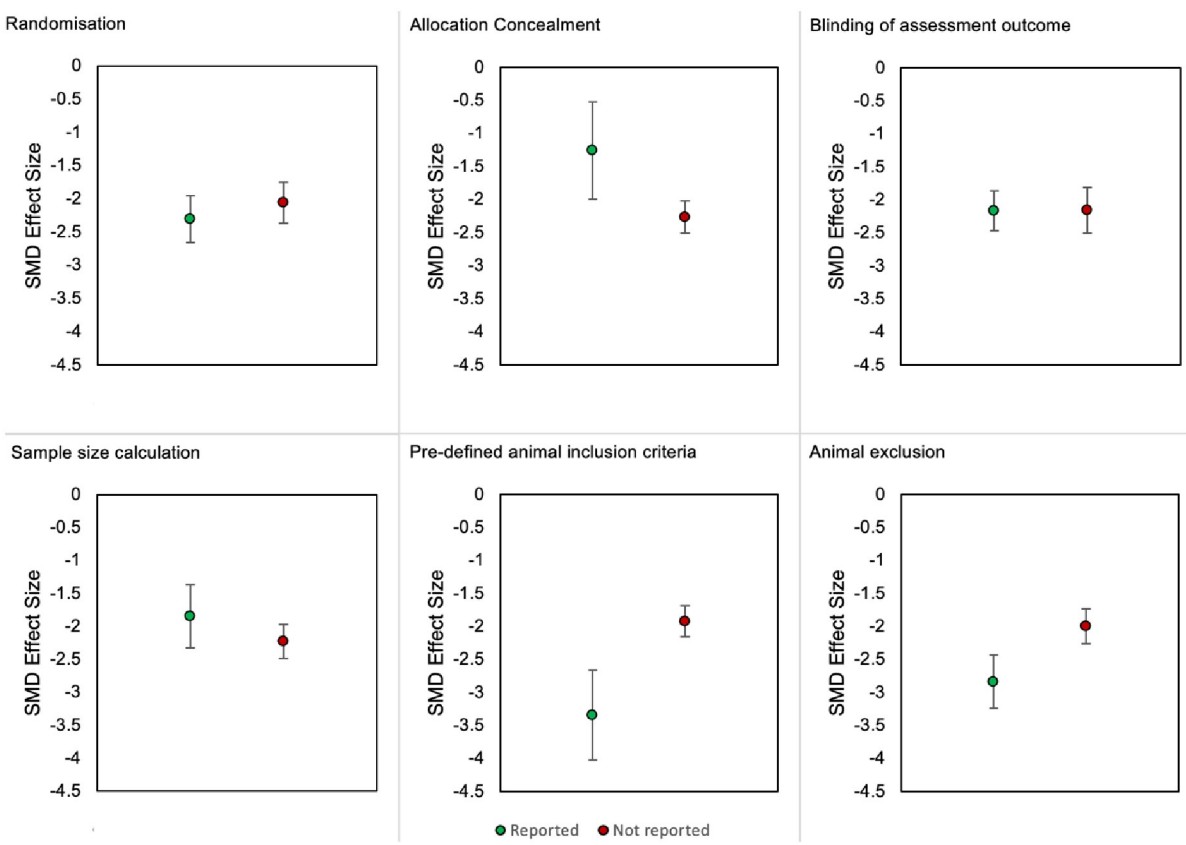

**Fig 26. Thigmotactic effect sizes associated with the reporting of the six methodological quality criteria in animal modelling experiments of rats and mice.** SMD, standardised mean difference.

of timings that the OFTs were conducted and limited comparisons within each period, we were also unable to discern how long after model induction it takes for thigmotactic behaviour to develop. Our findings are similar to those reported in a recent systematic review of mouse behavioural tests of anxiety, where the authors found a high-level heterogeneity in the data, as well as a high level of between-study variation in studies which used OFTs [31]. Future experimentation with increased biological variability and improved reporting will allow for sources of heterogeneity to be re-assessed.

We were unable to ascertain the effect of drug class on thigmotactic outcomes because there are many novel compounds that were only assessed within a single study. Gabapentinoid was the most investigated drug class in rat experiments and was mostly assessed in neuropathy models. This is expected as gabapentinoids are clinically used to treat neuropathic pain, although they have mixed efficacies depending on the type of neuropathy [32–34]. In mouse experiments, a hydrogen sulphide donor was the most investigated drug class and was mostly assessed in sciatic nerve ligation models. Preclinical studies have demonstrated the antinociceptive effects of hydrogen sulphide donors in arthropathy, inflammatory and neuropathy models [35–37]. Similarly, a naproxen-based hydrogen sulphide donor, has demonstrated analgesic efficacy in osteoarthritis patients [38, 39]. The broad range of drugs and drug classes has therefore not provided insight into which drugs could be used as comparators in studies investigating the efficacy of novel drugs in reducing thigmotaxis caused by injury and disease models relating to persistent pain.

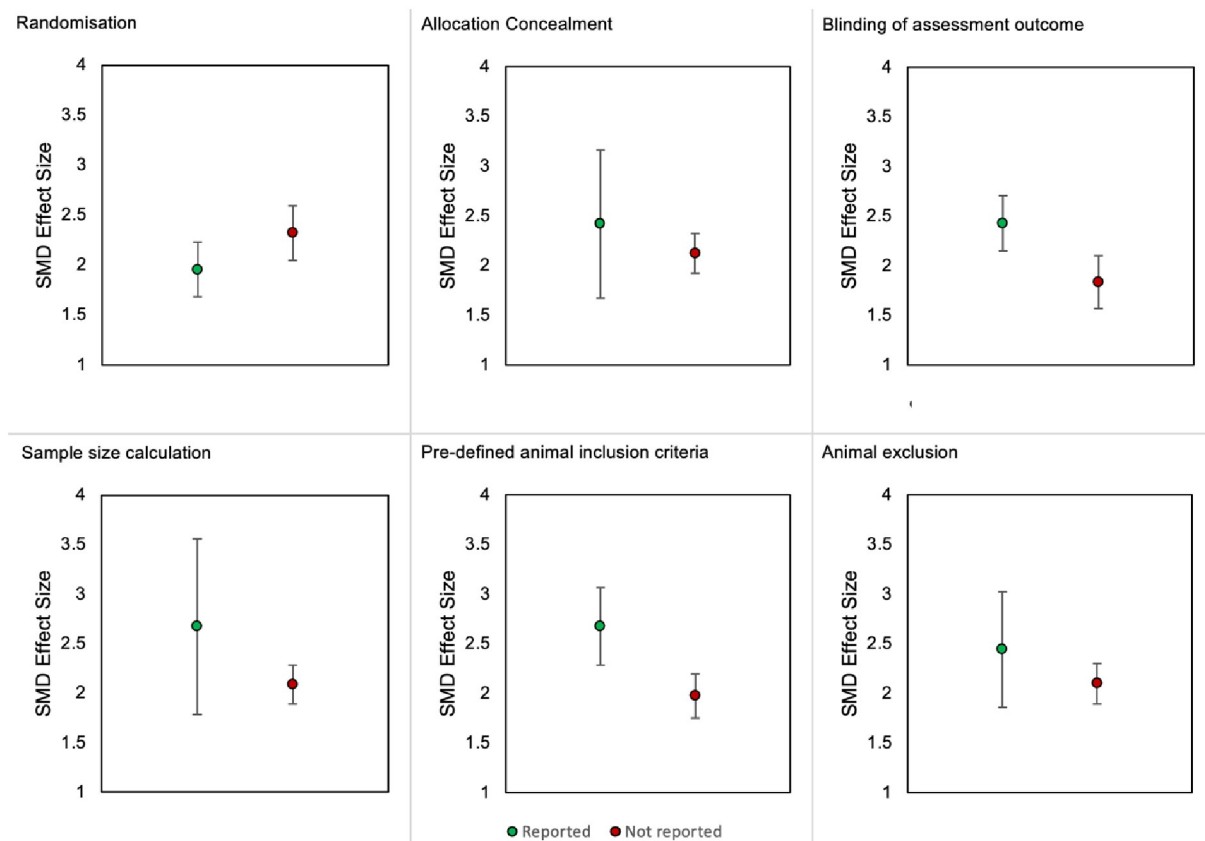

**Fig 27. Thigmotactic effect sizes associated with the reporting of the six methodological quality criteria in drug treatment experiments of rats and mice.** SMD, standardised mean difference.

We were also unable to ascertain differences in thigmotactic outcomes between strains because of the predominant use of Sprague-Dawley and Wistar rats and C57BL/6 mice. This issue of homogeneity in the rodent strain used for preclinical pain research has been observed in other systematic reviews [40–42]. Studies have shown that modification of behavioural outcomes caused by disease models associated with persistent pain is strain dependent [43–45]. Furthermore, the efficacy of an analgesic drug at the same dosage can vary between strains [46, 47]. We therefore advocate that future experiments have animals with diverse genetic profiles

**Table 3. Reporting of methodological quality criteria of the 22 studies which stated reporting in accordance with the ARRIVE guidelines but did not provide a checklist.**

| Methodological quality criteria | No. of studies reported | No. of studies that reported the method | No. of studies that scored low risk of bias | No. of studies that scored unclear risk of bias | No. of studies that scored high risk of bias |
|---|---|---|---|---|---|
| Random group allocation | 13 | 5 | 5 | 17 | 0 |
| Allocation concealment | 6 | 1 | 1 | 21 | 0 |
| Blinding of outcome assessment | 14 | 6 | 6 | 16 | 0 |
| Sample size calculation | 8 | 4 | 4 | 18 | 0 |
| Pre-defined animal inclusion criteria | 6 | 6 | 6 | 16 | 0 |

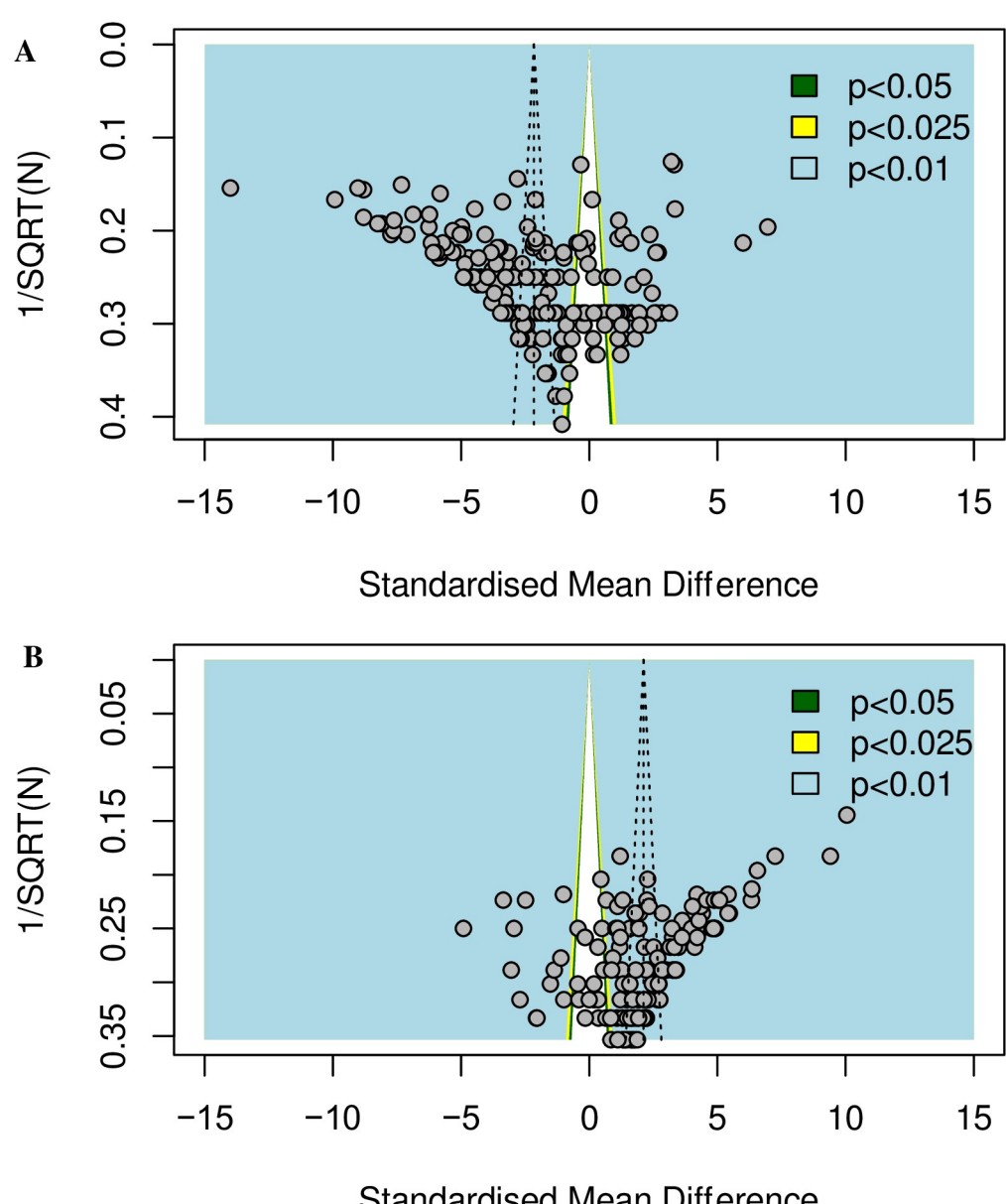

**Fig 28. Assessment of other biases in (A) animal modelling and (B) drug treatment experiments of rats and mice.**
The vertical dashed line represents the overall effect size. Filled circles represent experiments from the published studies. The coloured backgrounds indicate the statistical significance of effect sizes of cohort-level comparisons.

to improve the translatability and generalisability of the results to the heterogeneous clinical populations [3].

We could not discern the influence of sex on thigmotactic outcomes because male animals were predominantly used. In other preclinical systematic reviews of pain, sex accounted for a significant proportion of the heterogeneity [40–42, 48]. Women are clinically prone to be affected by chronic pain and tend to experience greater pain intensity than men [8]. Similarly, female rodents have shown to exhibit more hypersensitivity and nociceptive behaviours after injury [49–54]. In addition to differences in sensitivity, studies have shown sex dependent

analgesic efficacy [55–57]. To improve research translatability and generalisability, researchers need to maintain a sex balance in preclinical pain research, which is in line with funding bodies such as the National Institutes of Health policy [58], Canadian Institute of Health Research [59] and UK Research and Innovation [60].

**3.1.1. Thigmotactic outcome metrics.** Thigmotaxis can be reported by different outcome metrics within a study, and we employed our own hierarchy to extract the highest-ranking thigmotactic outcome metric when multiple metrics were reported (Table 3). The relationship between types of thigmotactic metrics and effect sizes remains uncertain due to insufficient data. Time spent in the centre is the most frequently reported thigmotactic metric and has shown sensitivity to disease models and drug treatments.

Injury or disease models can affect locomotor function, but it is unclear how different thigmotactic metrics are impacted by motor deficits. Total distance travelled by an animal can serve as a confounding outcome for detecting motor disturbance caused by an intervention. We assessed the correlation between each type of thigmotactic metric and the total distance travelled. We found a positive correlation between time spent in the centre and total distance travelled in rat experiments, indicating that thigmotaxis is associated with the locomotor ability and may be limited as an outcome measure in rats. We recommend that researchers use the "time spent in the centre/periphery" metric when assessing thigmotaxis because theoretically this metric has the lowest reliance on the locomotor function, as opposed to metrics like entries to the centre, number of central crossings and distance travelled in the centre. We also suggest researchers also assess total distance travelled and consider it when interpreting the results. There are other parameters that can be measured during the OFT which do not rely on locomotor function. Self-grooming behaviour is also an ethologically relevant behaviour and that has been increasingly employed in neuroscience research [61] but is still relatively novel in rodent pain research [62, 63].

**3.1.2. Experimental conditions and OF apparatus.** We looked at variables relating to experimental conditions and OF apparatus, however, due to the poor reporting, we could not draw meaningful conclusions on the associations between experimental design, OF apparatus and thigmotactic outcomes. There is a limited amount of research exploring the effect OF apparatus and experimental conditions on thigmotactic outcomes. A study by Eilam [64] reported that total distance and locomotion time were not dependent on the arena dimensions, while the spatial distribution of activity varied with arena size. Animals moved across the entire arena relatively evenly in smaller arenas, whereas they preferred to move near the walls in larger arenas. Another study found that centre horizontal activity was significantly greater in a circular-shaped arena as opposed to a square-shaped arena [65]. Experimental lighting is a crucial variable to control during the OFT because thigmotactic behaviour is enhanced in rodents that are exposed to high light intensity [66]. Current evidence for the effect of circadian phase on thigmotaxis during the OFT is conflicting [67, 68]. The length of the habituation to the OF and repeated OF testings using the same animals may also influence thigmotactic outcomes. Grabovskaya and Salvha [65] observed reduced thigmotaxis in animals that participated in repeated testing. Improved reporting of experimental conditions including apparatus used will improve our understanding of what experimental variables can influence thigmotactic outcomes and the reproducibility of results.

## 3.2. Correlation of thigmotaxis and stimulus-evoked limb withdrawal

Our findings are inconsistent: we identified a positive correlation between time in the centre and mechanical-induced behavioural outcomes in nerve-injured rats, rat analgesic drug experiments and mouse modelling It is likely that correlation will depend on the type of model

induced. However, it may not be appropriate to look for correlations between thigmotaxis and stimulus-evoked limb withdrawal because they capture different aspects of the pain experience. We suggest that researchers conduct a battery of tests comprised of both stimulus-evoked and ethologically relevant behavioural outcomes to improve validity and maximise the information gained from preclinical pain research.

### 3.3. Risk of bias

**3.3.1. Internal validity.** We could not accurately assess the internal validity of the included studies because the reporting of methodological quality measures to reduce the risk of bias was low. Most studies therefore have an unclear risk of bias. The infrequent reporting of blinding, including allocation concealment, could be a result of an inability to perform blinding due to obvious phenotypic or behavioural differences, or the belief that it is not necessary given the fact that thigmotaxis is an objective measure. Our analysis consistently identified that the reporting of pre-defined animal inclusion criteria was associated with larger effect sizes in both animal modelling and drug treatment experiments. Larger effect sizes were also associated with the reporting of animal exclusion in animal modelling experiments and the reporting of blinding in drug treatment experiments. In contrast, smaller effect sizes were associated with the reporting of allocation concealment in animal modelling experiments. It was not possible to accurately assess the risk of bias because the methods used to mitigate bias were also rarely reported, therefore the validity of these studies remains unclear. Similar observations were also found in other preclinical systematic reviews of pain [40–42, 48], suggesting that this is a general problem. The ARRIVE guidelines [69] were developed to improve the reporting of animal research and despite their introduction in 2010, and yet most of the post-2010 studies did not report in accordance with them. For studies which stated reporting in accordance with the ARRIVE guidelines, checklists were rarely provided. This lack of convention in the preclinical field limits our ability to assess reliability of the research findings. Researchers should use tools such as the EQIPD quality system [70] to design methodologically robust experiments and report in accordance with an established guideline such as ARRIVE.

**3.3.2. Other biases.** Our funnel plot asymmetry analysis suggests that other biases, including publication bias, may be present in both animal modelling and analgesic drug experiments. However, it should be noted that funnel plot asymmetry can be also caused by other factors such as study quality and selective reporting bias, where studies are published but outcomes with results that were not statistically significant [71]. Moreover, trim-and-fill analysis is pertained to generate unreliable results when the between-study heterogeneity is large [72–74], which may explain why trim-and-fill did not impute any missing experiments despite a significant Egger's regression

### 3.4. Limitations

To retrieve studies reporting OFTs and thigmotaxis, we conducted searches on multiple databases using a search strategy that had a good balance between sensitivity and specificity. However, there is always a possibility that not all relevant studies have been identified by the search which may affect the quality of the systematic review. But since the searches were all conducted systematically, the included studies represent an unbiased sample.

We can only rely on what has been reported in studies. It is difficult to evaluate risk of bias because potentially methods used to mitigate bias were implemented but not reported; conversely, methodological quality measures may have been reported but not performed. Similar observations have also been found in other preclinical systematic reviews of pain [40–42, 48].

In our search, there were 16 studies that met the inclusion criteria but could not be included in the meta-analysis due to not reporting key information, i.e., variance, sample size and thigmotactic data. Given the small sample sizes of the studies that were excluded from the meta-analysis, we think it is unlikely that the overall conclusion would change if that missing information is later provided.

We could not compare different animal characteristics (i.e., strain, sex, and drug class) within the same study type because of limited data. It was also not possible to draw meaningful conclusions on the associations between thigmotactic outcomes, study design characteristics, experimental conditions and OF apparatus because they are not reported frequently or in sufficient detail. There are other variables and factors that may influence thigmotaxis, such as sedative effects of a drug which were outside of the scope of this review but similarly are likely to suffer from poor reporting. Findings of this review should therefore be interpreted with caution and the impact of these variables on thigmotaxis need to be confirmed when more data become available and through prospective experiments. Until then, researchers should still carefully control these variables and transparently report their experimental designs.

In this review, we included any studies that assessed the treatment effect of a drug intervention on thigmotaxis by administering it to animals with injury and disease models relating to persistent pain. Most of the investigated drugs were novel compounds from a single study and many of them may never be developed for clinical trials. Additionally, we did not determine the rationale for the study, but future reviews should consider whether a study's use is mechanistic or designed to assess efficacy.

We only extracted data from the highest-ranking thigmotactic metric according to our hierarchical system when multiple metrics were reported within a study. As a result of this criterion, thigmotactic data included in the meta-analysis were mostly measured by time spent in the centre. Due to the insufficient number of data measured by metrics except the highest-ranking "time spent in the centre" metric, we were unable to conduct analyses to determine whether type of metrics used can be a source of heterogeneity and the correlation of these metrics with total distance travelled and stimulus-evoked limb withdrawal.

We decided to extract behavioural data at the time point at which there was the largest difference between control and treatment animals. This allowed us to calculate treatment effects independent of their treatment duration. Although we extracted the following information relating to the timing of the treatment: the time between model induction and the first or last OFT, how long before or after the model was induced was the first dose administered, and how long after the treatment started was the first OFT. Due to the large between-study variation and a low number of cohort-level comparisons, we could not investigate the impact of different treatment timings.

Lastly in our meta-analysis, we grouped together injury and disease models by type, however they may share different underlying aetiologies. Likewise, drugs were grouped according to their shared mechanism of action regardless of their other properties.

## 4. Conclusion

Given the translational challenges and unmet medical need to understand the underlying mechanisms and identify treatments for persistent pain, it is important to use animal models and outcome measures that have clinical relevance and ethological validity. This systematic review and meta-analysis provides a comprehensive summary of studies that investigated the effect of injuries and disease models associated with persistent pain and analgesic drug treatments on rodent thigmotactic behaviour. Its use, however, may be limited in certain injury and disease models because our analysis suggested that thigmotaxis may be associated with

locomotor function. We have identified where the external validity of preclinical pain research can be improved through increased biological variation e.g., female animals, more genetically diverse strains. To improve internal validity and assessment of reliability of results researchers should mitigate experimental biases and report their methods transparently. In the quest for clinically relevant outcomes, stimulus-evoked and ethologically relevant behavioural paradigms should be viewed as two separate entities because they are conceptually and methodologically different from each other that can be used and interpreted together.

## 5. Materials and methods

The protocol was registered on PROSPERO (CRD42040408044 https://www.crd.york.ac.uk/prospero/display_record.php?RecordID=208044) and published [75].

### 5.1. Search strategy

We systematically searched Ovid EMBASE, PubMed, and Web of Science on 8 October 2020, 30 March 2022 and 2August 2022 with no restrictions on languages and date of publication. The full search strategy for each database is provided in S5 File. Studies were amalgamated into an Endnote library (version 20.4) and duplicates removed.

### 5.2. Eligibility criteria

Inclusion criteria:

- Population: *in vivo* rodent models of disease associated with persistent pain

- Intervention: any clinically approved or novel drug analgesics used to interfere with nociception

- Comparison:–a cohort of control animals

  - For animal modelling experiments, a control population was defined as sham or naïve. For studies which used transgenic rodents to study persistent pain, a wild-type control was required

  - For studies which investigated the effect of analgesic drug interventions on rodent thigmotaxis, a vehicle control was required

- Outcome: thigmotactic behavioural metrics assessed during the OFT, which are:

  - Time spent in the centre

  - Time spent in the periphery

  - Entries to the centre

  - Latency to enter the centre

  - Number of central crossings

  - Distance travelled in the centre

  - Distance travelled in the periphery

For the meta-analysis, a study was required to report the following data: 1) the mean thigmotactic outcome, 2) its variance (i.e., standard deviation (SD) or standard error of the mean (SEM)) and 3) the number of animals per treatment group.

Exclusion criteria:

Non-rodent *in vivo* studies. Studies that investigated acute nociception. Studies that did not investigate thigmotactic outcome in the OFT. All other control conditions (i.e., baseline measurements) and/or absence of a control group. Non-primary research articles.

## 5.3. Study selection

Retrieved studies were screened on the Systematic Review Facility (SyRF) [76]. Studies were screened against the inclusion criteria in duplicate based on 1) titles and abstracts (TiAb), and 2) full texts. Since thigmotaxis can be referred to in numerous ways and sometimes information relating to behavioural assessments is not reported in detail in abstracts, we applied a strategy of study overinclusion (i.e., included studies that reported pain-associated rodent behaviours in the abstracts) during the title abstract screening to prevent the risk of falsely omitting relevant studies. Screening was completed by at least two independent reviewers and discrepancies were resolved by a third independent reviewer.

## 5.4. Data extraction

Data extraction was conducted on SyRF by two independent reviewers and discrepancies reconciled by a third independent reviewer.

For qualitative analysis, study-level data were extracted including bibliographic data, reporting quality, animal husbandry, and OFT characteristics (Table 5.1 in S5 File). Studies that were eligible for quantitative meta-analysis had experimental design and data extracted included animal model characteristics, intervention details and outcome measures(Table 5.2 in S5 File).

The primary outcome of interest was an outcome metric that denoted thigmotaxis. Thigmotaxis can be measured differently during an OFT and often multiple thigmotactic outcome metrics are reported within a study. Therefore, a hierarchy was applied (Table 4) when multiple metrics were reported. Reviewers extracted the data of the highest-ranking thigmotactic outcome metric. The other thigmotactic outcomes reported were noted. Metrics were ranked based on their relevance to the definition of thigmotaxis (i.e., decreasing time spent in the aversive open arena and/or increasing time spent in in proximity of the sheltered wall area). Total distance travelled in the OFT was also extracted because locomotor activity can be impaired by the interventions, so it serves as a confounding outcome. The secondary outcome was any stimulus-evoked limb withdrawal when assessed in the same cohort of animals used in the OFT.

Continuous data were extracted independent of the unit of measurement. Digital ruler software (Webplotdigitizer) was used to manually extract graphically presented data. When multiple time points were reported, the time point of the maximum effect was extracted. If the type

**Table 4. Thigmotactic outcome metrics hierarchy.**

| Hierarchical System |
| --- |
| 1) Time in the centre |
| 2) Time in the periphery (obverse to time in the centre) |
| 3) Entries to the centre |
| 4) Latency to the centre |
| 5) Number of central crossings |
| 6) Distance travelled in the centre |
| 7) Distance travelled in the periphery (obverse to distance travelled in the centre) |

of variance (i.e., SD or SEM) was not reported, it was characterised as SEM (i.e., to give the most conservative estimate). The most conservative estimate was extracted when data (e.g., sample size) were given as a range. When key information was unclear or not reported (i.e. mean data, variance, sample size), the corresponding authors were contacted. If the author did not respond or could not provide the information, the study was recorded as having missing data and was excluded from the meta-analysis.

## 5.5. Risk of bias assessment

An adapted version of the CAMARADES checklist and SYRCLE Risk of Bias tool [77, 78] were used to assess the reporting of six methodological quality criteria: random group allocation, allocation concealment, blinding of outcome assessment, sample size calculation, redefined animal inclusion criteria and animal exclusions. Reviewers stated whether each criterion was reported and included the description of the method used. Each criterion was rated separately according to the following criteria: low risk (accepted methods and were adequately described), high risk (inappropriate methods that did not efficiently mitigate bias), and unclear risk (the methodological quality criterion was not reported, or details of methods were insufficiently reported). Reporting of conflict of interests and compliance of animal welfare regulations were also extracted but were not included in the overall risk of bias.

## 5.6. Reconciliation

The data extracted by 2 independent reviewers were compared and any discrepancies reconciled by a third independent reviewer. For graphically presented data, the standardised mean difference effect sizes of individual comparisons were calculated for each reviewer's extracted data. When individual comparisons differed by <10%, the reconciler took an average of the two means and variance measures. When they differed by >10%, the reconciler had to extract the outcome data.

## 5.7. Data analysis

Thigmotactic data were separated by species and the type of experiments (i.e., animal modelling or drug experiments). The data is therefore divided into 4 datasets: (i) modelling of persistent pain in rats (ii) effects of drug interventions in rat models of persistent pain (iii) modelling of persistent pain in mice (iv) effects of drug interventions in mouse models of persistent pain. For animal modelling experiments, data were further separated according to the type of model control (sham or naïve) and were analysed separately. The number of independent cohort-level comparisons (k) required for each meta-analysis is ≥10. When k is <10 a descriptive summary was presented. Subgroup analyses were conducted to investigate how study characteristics influence effect sizes. All analyses were conducted using R statistical packages: dmetar (version 0.0.9); meta (version 6.0.0); and metafor (version 3.8.1).

**5.7.1. Effect size calculation.** For every individual comparison (defined as a cohort of animals that received an intervention compared to a control group), an effect size was calculated using the Hedges' g standardised mean difference (SMD) method. To obtain the "true number of control animals", all sample sizes was corrected by dividing the reported number of animals in the control group by the number of treatment groups it served. Effect sizes were weighted using the inverse variance method to reflect the contribution of each comparison to the total effect estimate. Stimulus-evoked limb withdrawal behavioural data were grouped according to the type of stimulus: mechanical, heat and cold. When more than one outcome of the same type of stimulus was reported from the same cohort of animals, a single nested effect size, which denotes a summary effect of the cohort, was calculated throughout. Cohort-level effect

sizes were pooled using a random-effects model as it considers within-study and between-study variances. The restricted maximum-likelihood method was used to estimate the variance of the distribution of true effect sizes [79]. The Hartung-Knapp-Sidik-Jonkman method was also applied to adjust confidence intervals [80–82].

**5.7.2. Heterogeneity.** Cochran's Q and $I^2$ tests were conducted. A $p$ value was calculated for Q, indicating whether all cohort-level comparisons shared a common effect size ($p > 0.05$) or not ($p < 0.05$). $I^2$ test calculated the proportion of total variance between studies that is due to true differences in effect sizes as opposed to chance. $I^2$ values were interpreted according to the definition given by Higgins and Thompsons [83]: 0–25% indicates very low heterogeneity; 25–50% indicates low heterogeneity; 50–75% indicates moderate heterogeneity; and >75% indicates high heterogeneity.

**5.7.3. Subgroup analyses.** Stratified meta-analyses for categorical variables were performed according to rodent species, strain, sex, model class, drug class, type of thigmotactic outcome metric, time (rat and mouse nerve injury models) and methodological quality criteria. Multiple meta-regressions were conducted to identify other variables relating to experimental conditions and OF apparatus that could influence the thigmotactic outcome (Table 5.3 in S5 File for detailed list).

**5.7.4. Other biases.** Funnel plots were generated to visually inspect plot asymmetry. SMDs were plotted against sample size-based precision estimate ($1/\sqrt{N}$) [84]. Egger's regression test provided a statistical assessment of the presence of biases including publication bias. Trim-and-fill analysis attempted to correct funnel plot asymmetry by imputing the theoretically missing studies and enabled a recalculation of the effect size.

**5.7.5. Correlation of thigmotaxis and total distance travelled in the OFT.** Cohort-level comparisons of experiments that assessed both thigmotaxis and total distance travelled were used to investigate correlation. A Pearson's correlation test was conducted between total distance travelled and each thigmotactic metric type. Regardless the number of cohort-level comparisons, post-hoc analyses were also conducted by pooling SMDs of total distance travelled from animals that received the same treatment (i.e. the same model and drug class) using a random-effects model accompanied by the restricted maximum-likelihood method and the Hartung-Knapp-Sidik-Jonkman method.

**5.7.6. Correlation of thigmotaxis and stimulus-evoked limb withdrawal.** Cohort-level comparisons of experiments that assessed both thigmotaxis and stimulus-evoked limb withdrawal were used to investigate correlation. A Pearson's correlation test was conducted between each thigmotactic metric type and each stimulus type (mechanical, heat, cold).

## Supporting information

**S1 File. Summary tables of acclimatisation, animal husbandry and experimental conditions.**
(DOCX)

**S2 File. Analysis of experiments using naïve controls.**
(DOCX)

**S3 File. Effects of experimental conditions and apparatus on thigmotactic outcomes.**
(DOCX)

**S4 File. Traffic light plot for individual study risk of bias.**
(XLSX)

**S5 File. Search strategy and data extraction criteria.**
(DOCX)

**S1 Checklist. PRISMA 2020 checklist.**
(DOCX)

## Acknowledgments

The authors thanks Daniel Corcoran and Nicolai van der Steen who contributed to the data extraction.

## Author Contributions

**Conceptualization:** Jan Vollert, Emily Sena, Andrew S. Rice, Nadia Soliman.

**Data curation:** Xue Ying Zhang, Marta Diaz-delCastillo, Lingsi Kong, Natasha Daniels, William MacIntosh-Smith, Aya Abdallah, Dominik Domanski, Denis Sofrenovic, Tsz Pui (Skel) Yeung, Diego Valiente.

**Formal analysis:** Xue Ying Zhang.

**Funding acquisition:** Andrew S. Rice.

**Investigation:** Xue Ying Zhang, Marta Diaz-delCastillo, Lingsi Kong, Natasha Daniels, William MacIntosh-Smith, Aya Abdallah, Dominik Domanski, Denis Sofrenovic, Tsz Pui (Skel) Yeung, Diego Valiente, Nadia Soliman.

**Methodology:** Xue Ying Zhang, Jan Vollert, Andrew S. Rice, Nadia Soliman.

**Project administration:** Nadia Soliman.

**Supervision:** Jan Vollert, Nadia Soliman.

**Visualization:** Xue Ying Zhang.

**Writing – original draft:** Xue Ying Zhang, Lingsi Kong, Natasha Daniels, William MacIntosh-Smith, Aya Abdallah, Dominik Domanski, Denis Sofrenovic, Tsz Pui (Skel) Yeung, Diego Valiente, Nadia Soliman.

**Writing – review & editing:** Xue Ying Zhang, Marta Diaz-delCastillo, Lingsi Kong, Natasha Daniels, William MacIntosh-Smith, Aya Abdallah, Dominik Domanski, Denis Sofrenovic, Tsz Pui (Skel) Yeung, Diego Valiente, Jan Vollert, Emily Sena, Andrew S. Rice, Nadia Soliman.

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
