## [Decision Letter · Decision Letter 0]

15 May 2023

PONE-D-23-05023A systematic review and meta-analysis of thigmotactic behaviour in the open field test in rodent models associated with persistent painPLOS ONE

Dear Dr. Soliman,

Thank you for submitting your manuscript to PLOS ONE. After careful consideration, we feel that it has merit but does not fully meet PLOS ONE’s publication criteria as it currently stands. Therefore, we invite you to submit a revised version of the manuscript that addresses the points raised during the review process.

ACADEMIC EDITOR: - please do follow the suggestion provided by reviewers - have in mind that some of the data could be relocated to supplementary files.- do support your study introduction and discussion with relevant and the most recent reviews and studies

We look forward to receiving your revised manuscript.

Kind regards,

Dragan Hrncic

Academic Editor

PLOS ONE

Journal Requirements:

"The authors thanks Daniel Corcoran and Nicolai van der Steen who contributed to the data extraction. Zhang XY thanks the funding support from the Medical Research Council Doctoral Training Partnership Programme (grant number MR/N014103/1). Soliman is funded by the Jennie Gwynn legacy fund. "

5. We note that you have stated that you will provide repository information for your data at acceptance. Should your manuscript be accepted for publication, we will hold it until you provide the relevant accession numbers or DOIs necessary to access your data. If you wish to make changes to your Data Availability statement, please describe these changes in your cover letter and we will update your Data Availability statement to reflect the information you provide."

6. Please amend the manuscript submission data (via Edit Submission) to include author "Kong L, Soliman N, Daniels NF, Macinotash-Smith W, Abdallah A, Domanski D, Sofrenovic D, Yeung TP, Valient J., Rice ASC

Reviewers' comments:

Reviewer's Responses to Questions

**Comments to the Author**

1. Is the manuscript technically sound, and do the data support the conclusions?

Reviewer #1: Yes

2. Has the statistical analysis been performed appropriately and rigorously? 

Reviewer #1: Yes

3. Have the authors made all data underlying the findings in their manuscript fully available?

Reviewer #1: Yes

4. Is the manuscript presented in an intelligible fashion and written in standard English?

Reviewer #1: Yes

5. Review Comments to the Author

Reviewer #1: This manuscript by Zhang and colleagues is a systematic review and meta-analysis of thigmotaxis in models of pain. I am impressed by the still rather novel attempt to meta-analyze preclinical data, although this paper again shows why it doesn’t really show much: simply not enough variability in methods (e.g., use of different sexes, strains, multiple drugs). I’m also a bit underwhelmed by the stakes here; 30+ figures and long, overly detailed tables in service of what most people in the field view as a rather secondary issue (pain comorbidities, or potential confounds). That said, I realize PLoS One is not concerned about how important this all is, but rather whether it’s solid. It clearly is. This group is absolutely at the forefront of preclinical pain meta-analysis. I have two major pieces of advice to improve things.

The first is that to my mind the group failed to analyze what I think is probably the most important issue: time. They conclude that thigmotaxis is caused by persistent pain. Um, duh. We knew this. The outstanding question in the field is how long does it take for the anxiety (which is driving the thigmotaxis) to develop after the injury. This information (time of anxiety testing post-injury) is in every paper they analyzed, but I see no evidence that it was looked at here.

The second is that I don’t like the introduction at all. The citations are utterly arbitrary, in some cases randomly picking a few papers when much better reviews exist, and in others using arbitrarily chosen reviews instead of citing primary literature proving a point. But the bigger problem is that the discussion is about predictive validity of outcome measures of pain, when the paper is actually meta-analyzing not an outcome measure of pain, but rather a measure of a pain-induced comorbidity: anxiety. No one in the preclinical pain field is pretending that thigmotaxis is a measure of pain itself, however indirect. They are using it specifically to address the presence or absence of the comorbidity. Thus, I think this paper would be a lot better set up by a discussion of the prevalence of pain comorbidities and their modelling in animals.

6. PLOS authors have the option to publish the peer review history of their article (what does this mean?). If published, this will include your full peer review and any attached files.

Reviewer #1: No

---

## [Author Response · Author response to Decision Letter 0]

28 Jul 2023

Reviewer #1: This manuscript by Zhang and colleagues is a systematic review and meta-analysis of thigmotaxis in models of pain. I am impressed by the still rather novel attempt to meta-analyze preclinical data, although this paper again shows why it doesn’t really show much: simply not enough variability in methods (e.g., use of different sexes, strains, multiple drugs). I’m also a bit underwhelmed by the stakes here; 30+ figures and long, overly detailed tables in service of what most people in the field view as a rather secondary issue (pain comorbidities, or potential confounds). That said, I realize PLoS One is not concerned about how important this all is, but rather whether it’s solid. It clearly is. This group is absolutely at the forefront of preclinical pain meta-analysis. I have two major pieces of advice to improve things.

We share the reviewer’s underwhelm and agree that this review hasn’t led to the insights one would hope for to progress the field. There remains an ethical and societal obligation to continue to examine the use of animal models in pain research and given the volume and exponential production of data, systematic reviews are likely to play an important role. However, the lack of variability in experimental designs and the paucity of reporting experimental details are limiting. We know pain studies are affected by a range of factors including methodological rigour, animal characteristics, experimental paradigms and the environment, however, as this review demonstrates, we are yet to see the changes in the field that would improve our appreciation of these factors. 

The first is that to my mind the group failed to analyze what I think is probably the most important issue: time. They conclude that thigmotaxis is caused by persistent pain. Um, duh. We knew this. The outstanding question in the field is how long does it take for the anxiety (which is driving the thigmotaxis) to develop after the injury. This information (time of anxiety testing post-injury) is in every paper they analyzed, but I see no evidence that it was looked at here.

We collected time from model intervention to open field test and have conducted an analysis of time in rat and mouse nerve injury models. The methods, results and discussion have been changed to reflect this. The analysis was performed on rat and mouse nerve injury models (the most frequently reported) and are presented in figures 7 and 20. 

Results section 2.3.1., “Open field tests were conducted from immediately after model induction to 35 days after nerve injury (for which there were 41 comparisons that reported time since model induction). Time accounted for a significant proportion of heterogeneity (Q = 652.50; d.f. = 13; p <0.0001) (Figure 7A). However, this was not the case when time was analysed in weekly blocks (Q = 3.31; d.f. = 3; p =0.35) (Figure 7B).”

Results section 2.3.2., “Open field tests were conducted between 24 h and 180 days after nerve injury (for which there were 46 comparisons that reported time since model induction). All time points accounted for a significant proportion of heterogeneity (Q = 634.39; d.f. = 19; p <0.0001) (Figure 20A). The same was observed in the time block analysis (Q = 16.61; d.f. = 3; p =0.0009) and the 2-3 week time block resulted in the largest observed effect sizes (Figure 20B). However, these findings should be interpreted with caution given the low number of comparisons for each time period.”

Discussion, “Given the broad range of timings that the OFTs were conducted and limited comparisons within each period, we were also unable to discern how long after model induction it takes for thigmotactic behaviour to develop.”

The second is that I don’t like the introduction at all. The citations are utterly arbitrary, in some cases randomly picking a few papers when much better reviews exist, and in others using arbitrarily chosen reviews instead of citing primary literature proving a point. 

The introduction has been updated. We have included a broader range of more recent and relevant primary studies and reviews. This includes references to the systematic reviews Currie et al., 2019, Soliman et al., 2021 and narrative reviews Sadler et al., 2022, and Tappe-Theodor et al., 2014. 

But the bigger problem is that the discussion is about predictive validity of outcome measures of pain, when the paper is actually meta-analyzing not an outcome measure of pain, but rather a measure of a pain-induced comorbidity: anxiety. No one in the preclinical pain field is pretending that thigmotaxis is a measure of pain itself, however indirect. They are using it specifically to address the presence or absence of the comorbidity. Thus, I think this paper would be a lot better set up by a discussion of the prevalence of pain comorbidities and their modelling in animals.

We have described thigmotaxis as a behaviour not a measure of pain. Pain cannot be directly measured in humans or animals. We have contextualised thigmotaxis to pain-associated models and whether thigmotactic behaviour can be attenuated by analgesic drugs, thereby distinguishing between pain and anxiety. We appreciate that thigmotaxis can be influenced by other co-morbidities and this was discussed within the introduction e.g., “This behavioural paradigm is commonly used for animal research of psychiatric disorders, particularly anxiety…” and “It could also reflect the clinical observations of exacerbated avoiding behaviours and anxiodepressive disorders that are associated with patients with chronic pain”. Measuring an ethologically relevant behaviour allows us to measure the global affect of a pain-associated intervention. The following sentence has been added to the introduction to further emphasise the points made by the reviewer, “It has been postulated that the behaviour could give insight into the affective, cognitive and sensory dysfunction associated with pain.”

---

## [Decision Letter · Decision Letter 1]

8 Aug 2023

A systematic review and meta-analysis of thigmotactic behaviour in the open field test in rodent models associated with persistent pain

PONE-D-23-05023R1

Dear Dr. Soliman,

We’re pleased to inform you that your manuscript has been judged scientifically suitable for publication and will be formally accepted for publication once it meets all outstanding technical requirements.

Kind regards,

Prof. Dr. Dragan Hrncic,

Academic Editor

PLOS ONE

Additional Editor Comments (optional):

Reviewers' comments:

Reviewer's Responses to Questions

**Comments to the Author**

1. If the authors have adequately addressed your comments raised in a previous round of review and you feel that this manuscript is now acceptable for publication, you may indicate that here to bypass the “Comments to the Author” section, enter your conflict of interest statement in the “Confidential to Editor” section, and submit your "Accept" recommendation.

Reviewer #1: All comments have been addressed

2. Is the manuscript technically sound, and do the data support the conclusions?

Reviewer #1: (No Response)

3. Has the statistical analysis been performed appropriately and rigorously? 

Reviewer #1: (No Response)

4. Have the authors made all data underlying the findings in their manuscript fully available?

Reviewer #1: (No Response)

5. Is the manuscript presented in an intelligible fashion and written in standard English?

Reviewer #1: (No Response)

6. Review Comments to the Author

Reviewer #1: I am satisfied with the changes. I am satisfied with the changes. I am satisfied with the changes. I am satisfied with the changes. (Why is the minimum character count 100 in this field?!)

7. PLOS authors have the option to publish the peer review history of their article (what does this mean?). If published, this will include your full peer review and any attached files.

Reviewer #1: No

---

## [Editor Report · Acceptance letter]

30 Aug 2023

PONE-D-23-05023R1 

A systematic review and meta-analysis of thigmotactic behaviour in the open field test in rodent models associated with persistent pain 

Dear Dr. Soliman:

I'm pleased to inform you that your manuscript has been deemed suitable for publication in PLOS ONE. Congratulations! Your manuscript is now with our production department. 

Kind regards, 

on behalf of

Professor Dragan Hrncic 

Academic Editor

PLOS ONE